# Role of Aryl Hydrocarbon Receptor Activation and Autophagy in Psoriasis-Related Inflammation

**DOI:** 10.3390/ijms21062195

**Published:** 2020-03-22

**Authors:** Hye Ran Kim, Seok Young Kang, Hye One Kim, Chun Wook Park, Bo Young Chung

**Affiliations:** Department of Dermatology, Hallym University Kangnam Sacred Heart Hospital, Hallym University College of Medicine, Seoul 07441, Korea; cyberkhr@hanmail.net (H.R.K.); tjdjrdud@naver.com (S.Y.K.); hyeonekim@gmail.com (H.O.K.); dermap@hanmail.net (C.W.P.)

**Keywords:** aryl hydrocarbon receptor, autophagy, skin, psoriasis, 2,3,7,8-Tetrachlorodibenzo-p-dioxin (TCDD)

## Abstract

Aryl hydrocarbon receptor (AhR) and autophagy reportedly regulate immune responses in the skin. This study explored the effects of AhR activation on autophagy in human keratinocytes, and the relevance of AhR and autophagy in psoriasis pathogenesis. AhR activation by 2,3,7,8-Tetrachlorodibenzo-p-dioxin (TCDD) repressed autophagy, while autophagy inhibition induced AhR activation in HaCaT cells and normal human epidermal keratinocytes (NHEKs). A particularly strong interaction between AhR and autophagy was observed in proinflammatory cytokines-stimulated keratinocytes, an in vitro model of psoriasis. In skin biopsies from psoriasis patients, a similar impact of AhR on autophagy and inflammation was observed. AhR inhibition blocked TCDD- and chloroquine-induced p65NF-κB and p38MAPK phosphorylation in proinflammatory cytokines-stimulated HaCaT cells. Moreover, higher expression of AhR and CYP1A1, and lower expression of LC3, were detected in psoriatic skin tissues, compared to the controls. These data demonstrated that AhR modulated autophagy leads to skin inflammation in human keratinocytes via the p65NF-κB/p38MAPK signaling pathways, suggesting that AhR signaling and autophagy might be involved in the pathogenesis of chronic inflammatory disorders such as psoriasis.

## 1. Introduction

Aryl hydrocarbon receptor (AhR) is a ligand-dependent transcription factor that binds to exogenous and endogenous chemicals and modulates the expression of several genes, with positive or negative effects in various organs, including the skin [1]. Many ligands, such as vitamin D3 hydroxyderivatives can act on AhR [2]. Well-known ligands with high-affinity for AhR include several environmental contaminants, such as polycyclic aromatic hydrocarbons (e.g., 2,3,7,8-Tetrachlorodibenzo-p-dioxin (TCDD) and benzo[a]pyrene) [3]. TCDD is known as dioxin [4]. Dioxins are typically produced by garbage incineration and steel manufacturing, as well as a byproduct of herbicides and pesticides, including chlorophenols. Dioxins are also found in cigarette smoke. Due to their high lipophilicity and poor metabolism, dioxins are present in nearly all humans, with higher concentrations commonly found in individuals from industrialized countries [5]. TCDD has multiple biological and pathologic effects on humans even at low doses [6]. The major effects of TCDD are channeled by the binding and activation of intracellular AhR. [3]. Upon binding to ligands, cytoplasmic AhR translocates to the nucleus, dimerizes with AhR nuclear translocator (ARNT), and regulates a number of physiologic and pathological processes by inducing the expression of multiple AhR-responsive genes, such as the xenobiotic-metabolizing enzyme, CYP1A1 (cytochrome P450) [7]. Recent studies have demonstrated that AhR activation is related to innate immunity, epidermal barrier function, oxidation/antioxidation, photoinduced response, and melanogenesis [8,9,10,11]. 

Autophagy is a critical intracellular disintegrative process in which cytoplasmic factors are separated in double-membrane vesicles and decomposed by fusion with lysosomes. Autophagy plays an important role in maintaining cell homeostasis by removing dysfunctional or damaged organelles or proteins [12]. In addition, autophagy is known to regulate a variety of cell processes, e.g., cell apoptosis, pathogen removal, antigen presentation, and inflammation, and has been associated with a number of human disorders, including infectious and neurodegenerative diseases, cancer, and metabolic conditions [13]. Autophagy is controlled by the Atg (autophagy-related) family of proteins, among which the ubiquitin-like protein, microtubule-associated protein IA/IB light chain 3 (LC3), is involved in cytosolic cargo recruitment and autophagosome formation [14]. Hence, it is commonly used as an autophagy marker to indicate autophagosome quantity as the lipidated form of LC3 accumulates in autophagosomal membranes [15].

Psoriasis is a common, recurrent inflammatory chronic skin disease that results from mutual interactions between the environment and genetic factors. Psoriasis pathogenesis is believed to involve immune system abnormalities [16]. A previous study described an in vitro model based on inflammatory keratinocytes capable of reproducing lesional psoriatic skin. This model was based on keratinocyte stimulation with IL-17A, IL-22, Oncostatin-M, TNF-α, and IL-1α, which showed that the upregulation of chemokines and the production of antimicrobial peptides are related to psoriasis [17]. Recently, the roles of AhR and autophagy in innate and adaptive immunity have been demonstrated [18,19]. In fact, as a critical chemosensor, AhR activation regulates both innate and adaptive immune responses [20,21]. It was further identified that AhR affects the maturation and function of antigen-presenting on Langerhans cells [11]. AhR activation by TCDD and endogenous molecules also functions in the differentiation of T-helper 17 (Th17) and T regulatory (Treg) cells [22]. Additionally, autophagy also plays critical functions in modulating innate and adaptive immunity. In the innate immune response, autophagy not only contributes to the clearance of pathogens, but also participates in protection against toxins, and impacts cytokine production. In terms of the adaptive immunity, autophagy plays crucial roles in T cell maturation and differentiation, as well as antigen presentation [23]. However, the relationship between AhR and autophagy in the skin has not yet been elucidated. To the best of our knowledge, this study is the first to confirm the effects of AhR on autophagy and to describe a correlation between AhR and autophagy cross-talk and psoriasiform skin inflammation. 

To verify the impact of environmental factors on psoriasis, we focused on AhR effects on autophagy. To this end, we evaluated whether AhR activation regulated autophagy, or vice versa, in human keratinocytes under inflammatory conditions, as well as in skin biopsies from psoriasis patients, and explored the influence of AhR and autophagy on inflammation in skin biopsies from psoriasis patients. We further analyzed the expression levels of AhR, CYP1A1, and LC3 in psoriatic skin tissue.

## 2. Results

### 2.1. Cell Viability

To determine the cytotoxicity of TCDD, rapamycin, and chloroquine, 3-(4,5-dimethylthiazol-2yl)-2,5-diphenyl-tetrazolium bromide (MTT) assay was used. HaCaT cells and NHEKs were treated with different concentrations of TCDD (0, 0.1, 1, 10, and 100 nM), rapamycin (0, 50, 100, 150, and 200 nM) or chloroquine (0, 10, 20, 30, 40, and 50 µg/mL). No cytotoxicity was observed after treatment with TCDD concentrations between 0.1 and 10 nM, rapamycin concentrations up to 50 nM, or chloroquine concentrations as high as 30 µg/mL in both HaCaT cells and NHEKs. Although there was significantly reduced cell viability observed following treatment with 10 nM TCDD, 50 nM rapamycin, and 20 µg/mL chloroquine compared to unstimulated cells, the viability remained over 90% for all treatment conditions (Figure 1). Therefore, 10 nM TCDD, 50 nM rapamycin, and 20 µg/mL chloroquine were used for further experiments.

### 2.2. In M5-Stimulated HaCaT Cells, TCDD Treatment Significantly Upregulated AhR and CYP1A1 While Downregulating Autophagy-Related Factors and the Production of Autophagosomes 

We next examined the effect of TCDD in HaCaT cells, as well as in culture with a mixture of five proinflammatory cytokines, namely M5, that stimulate HaCaT cells, which exhibit the features of psoriasis (Appendix A). TCDD increased the expression of AhR and CYP1A1, with the effect being more pronounced in M5-stimulated HaCaT cells (Figure 2A). Moreover, TCDD decreased the expression of autophagy-related factors, including P62, ATG5, LC3, and Beclin 1, with a stronger downregulation observed in M5-stimulated HaCaT cells (Figure 2B). 

We also assessed formation of organelles displaying punctate LC3 staining (autophagosomes) by using endogenous LC3 as an autophagic marker. TCDD treatment decreased LC3 expression. This suppressive effect was more evident in M5-stimulated HaCaT cells as compared to that in unstimulated-HaCaT cells (Figure 2C,D).

### 2.3. Autophagy Inhibition Upregulated AhR and CYP1A1 in M5-Stimulated HaCaT Cells

Next, we assessed whether autophagy modulation affected the expression of AhR and CYP1A1, as well as LC3 gene and protein levels. Among the many autophagy-modulating drugs, chloroquine, an antimalarial drug, is a representative autophagy inhibitor that blocks the degradation of autophagosomes; while rapamycin is regarded as an autophagy inducer and inhibits mammalian TOR (mTOR), which is serine/threoine kinase [24,25]. Since autophagy is regulated by the mTOR complex, rapamycin can induce autophagy via inhibition of mTOR. Therefore, we selected rapamycin as an autophagy inducer and chloroquine as an autophagy inhibitor. HaCaT cells, with or without M5 stimulation, were treated with rapamycin (50 nM) or chloroquine (20 µg/mL). Chloroquine significantly upregulated AhR mRNA in HaCaT cells. A particularly pronounced upregulation of CYP1A1 and AhR mRNAs was observed in chloroquine+ M5-treated HaCaT cells (Figure 3A,B). Consistently, chloroquine increased AhR and CYP1A1 protein expression and decreased LC3 expression in M5-stimulated HaCaT cells, as assessed by Western blot (Figure 3C). As shown in Figure 3A,B, rapamycin did not induce AhR or CYP1A1 mRNA expression. Rather, a significant downregulation of CYP1A1 mRNA was observed in rapamycin + M5-treated HaCaT cells. Further, Western blot analysis revealed that rapamycin lead to a slight decrease in the level of AhR and CYP1A1 proteins as well as a slight increase in LC3 expression compared to the control (Figure 3C). 

To confirm the cross-talk between AhR and autophagy in human keratinocytes, we performed immunofluorescence staining in NHEKs with antibodies against AhR, CYP1A1 TNF-α, and LC3 after treatment with rapamycin, chloroquine, or TCDD. In NHEKs, the expression of AhR, CYP1A1, and TNF-α increased while the expression of LC3 significantly decreased with chloroquine or TCDD treatment (Figure 4A–E). Taken together, AhR and autophagy exhibited a reverse relationship in keratinocytes, and this phenomenon was particularly pronounced in the in vitro psoriasis model.

### 2.4. AhR Knockdown Significantly Attenuated Autophagy

To further examine the impact of AhR on autophagy in HaCaT cells, AhR blocking experiments were performed based on the use of siRNA oligonucleotides. All oligonucleotides significantly reduced the expression of AhR mRNA in non-stimulated conditions at 48 h post-transfection (72% reduction of AhR mRNA in siAhR-transfected cells relative to mock) (Figure 5A). HaCaT cells were transiently transfected with an AhR siRNA, followed by stimulation with TCDD (10 nM) and M5 (10 ng/mL) for 48 h. TCDD or M5 did not significantly affect the mRNA levels of ATG5, LC3, and Beclin1. AhR knockdown resulted in significant attenuation of the expression of autophagy-related factors relative to that in the respective controls (Figure 5B–D).

### 2.5. AhR Inhibition Blocked both TCDD- and Chloroquine-Induced Phosphorylation of p65NF-κB and p38MAPK in M5-Stimulated HaCaT Cells

To further understand the molecular basis of TCDD-induced activation of inflammatory signaling in HaCaT cells, we next investigated the effect of TCDD or chloroquine on the expression of p38MAPK and p65NF-κB in M5-stimulated HaCaT cells. Incubation of HaCaT cells with TCDD+M5 significantly induced p65NF-κB and p38MAPK phosphorylation, with a peak at 30 min (Appendix A). 

To ascertain whether the signaling pathways activated by TCDD and chloroquine were influenced by AhR in HaCaT cells, cells were transfected with siRNAs specifically blocking AhR expression, and subsequently treated with TCDD + M5 or chloroquine + M5 for 30 min. As shown in Figure 6A,B, TCDD treatment caused an increase in the level of phospho-p38 (p-p38) and p-p65, similarly to M5; while the M5/TCDD combinatorial treatment induced a stronger effect compared to the single treatments. Furthermore, chloroquine induced a slight increase in p38 and p65 phosphorylation. Similarly, M5 induced an increase in the level of p-p38 and p-p65. The combination of M5 and chloroquine also induced a stronger effect compared to the individual treatments (Figure 6C,D). AhR knockdown seems to not only attenuate the TCDD- or chloroquine (CQ)-mediated effects but also results in even stronger reductions in p65 and p38 phosphorylation, independently of TCDD or CQ. (Figure 6A–D).

### 2.6. CYP1A1, LC3, and AhR Differential Expression in Human Lesional Psoriasis Skin

The level of CYP1A1, AhR, and LC3 expression was determined by immunohistochemistry, and qPCR in human healthy skin (control) and human psoriasis lesion skin (Figure 7 and Figure 8). In human psoriasis lesion skin, the expression level of CYP1A1 and AhR was increased and LC3 was decreased compared to controls. 

### 2.7. AhR or Autophagy Modulation in Human Psoriasis Skin Biopsies Enhanced the Production of Proinflammatory Cytokines

To verify whether AhR or autophagy had a role in psoriasis pathology, we investigated the effects of TCDD-induced AhR activation, autophagy stimulation by rapamycin, or autophagy inhibition by chloroquine on the production of proinflammatory cytokines. Full-thickness skin biopsies were taken from lesion skin of psoriasis patients (*n* = 3) and healthy human volunteers (*n* = 3). Human skin explants were treated with rapamycin, chloroquine, or TCDD, and cultured for 48 h before mRNA quantification. AhR and CYP1A1 expression was significantly higher in TCDD-treated-psoriasis lesion skin tissues compared to TCDD-treated controls (Figure 9A,B). LC3 expression was significantly decreased in rapamycin- or TCDD-treated psoriasis lesion skin tissues, compared to rapamycin- or TCDD-treated controls (Figure 9C). Chloroquine or TCDD treatment of psoriasis skin biopsies enhanced the production of the proinflammatory cytokines, IL-1β, IL-6, and TNF-α compared to chloroquine- or TCDD- treated controls (Figure 9D–F). Thus, AhR and autophagy appeared to be critical factors in psoriasis-related skin inflammation. 

## 3. Discussion

In this study, we demonstrated that AhR negatively modulates autophagy in human keratinocytes. In particular, while TCDD-induced AhR activation caused autophagic repression, autophagy stimulation suppressed AhR activation in both HaCaT cells and NHEKs. M5-stimulated keratinocytes, an in vitro model of psoriasis, revealed a particularly strong relationship between AhR and autophagy. In skin biopsies from psoriasis patients, a similar relationship was observed, as well as cross-talk between AhR and autophagy induced inflammation. AhR-induced autophagy inhibition was dependent on NF-κB/p65 and mitogen-activated protein kinase (MAPK)/p38 signaling. In addition, psoriatic skin tissue exhibited higher expression levels of AhR and CYP1A1, and lower expression of LC3, compared to control tissues. 

The activation of AhR through TCDD in primary mouse keratinocytes enhances the expression of neutrophil-induced chemokine (C-X-C motif) ligand 5 (Cxcl5) [26]. Cxcl5 expression is upregulated in keratinocytes after IL-17 exposure, in association with psoriasis pathogenesis [27]. In mice with experimental autoimmune encephalomyelitis (EAE), AhR activation modulates both Treg and TH17 cell differentiation in a ligand-specific fashion [22]. Prolonged in vivo exposure to high doses of TCDD results in long-lasting effects on the adaptive immune system and differentiated CD4+ T cells, leading to IL-22 production [28]. IL-17 and IL-22 are known as crucial factors in the development of psoriasiform inflammation [29]. 

Autophagy has been regarded as an endogenous defense mechanism against environmental disturbances. Notably, in addition to maintaining skin homeostasis, autophagy has also been implicated in the development of skin disorders [30]. A recent study reported that single-nucleotide polymorphisms (SNPs) in the *ATG16L1* gene (rs10210302, rs12994971, rs2241880, rs2241879, and rs13005285) are linked to psoriasis susceptibility [31]. The ATG16L1 protein, a major component of the autophagy-related protein complex, is critical to the autophagy process [32]. Moreover, it has been speculated that an ATG16L1 defect affects the role of autophagy machinery in various signaling pathways involved in modulating cytokine production, resulting in the accumulation of dysfunctional proteins and organelles, and subsequent chronic inflammation [32]. Additionally, Lee et al. reported that autophagy deficiency in keratinocytes caused increased production of inflammatory cytokines, and cell proliferation via induction of the scaffolding adaptor protein p62/SQSTM1 (p62) expression [33]. However, studies on the role of autophagy in psoriasis pathogenesis remain limited. Both AhR signaling and autophagy act as skin homeostatic rheostats sensing environmental stimuli, and evidence suggests that AhR signaling and autophagy are involved in the pathogenesis of chronic inflammatory skin disease, such as psoriasis [22,31,33,34,35]. Thus, we suggested that AhR activation by environmental pollutants such as dioxin (TCDD) could affect the autophagic process, and might contribute to the pathogenesis of chronic inflammatory skin disorder, including psoriasis. 

As expected, TCDD treatment increased AhR and CYP1A1 expression in HaCaT cells and NEHKs. These effects were more evident in a cytokine-stimulated in vitro psoriasis model and in ex vivo psoriasis skin biopsies, compared to controls. These results suggested that AhR activation might be related to the development or aggravation of psoriasis. TCDD-induced AhR activation reduced the expression of autophagy-related proteins, as well as the production of autophagosomes. Moreover, autophagy inhibition by chloroquine increased AhR expression, at both mRNA and protein levels. Thus, in keratinocytes, an inverse correlation between AhR and autophagy was demonstrated. Such phenomenon was particularly obvious in an in vitro psoriasis model, as well in psoriasis skin biopsies. In human psoriasis skin lesions, the level of AhR and CYP1A1 proteins and transcripts was upregulated, while LC3 was downregulated at both mRNA and protein level, compared to controls. TCDD significantly increased the production of TNF-α, and inhibition of autophagy by chloroquine significantly increased the production of TNF-α in NHEKs. In psoriasis skin biopsies, the synthesis of proinflammatory cytokines, including IL-1β, IL-6, and TNF-α, was significantly enhanced by TCDD and chloroquine, compared to controls. We found that AhR-induced inhibition of autophagy was related to NF-κB/p65 and MAPK/p38 pathway signaling. To the best of our knowledge, this is the first demonstration of a reverse correlation between AhR and autophagy in skin. Based on our findings, we speculate that, in keratinocytes, the cross-talk between AhR and autophagy might trigger or boost chronic skin inflammation. 

The effects of AhR and autophagy on the skin remain unclear. Consistently with our study, autophagy inhibition may reflect proinflammatory response in the skin [33]. However, contrasting results also have been published. Jang et al. recently reported that exposure to particulate matter (PM) induced autophagy via AhR in HaCaT keratinocytes [36]. This discrepancy might be due to the different ligands used for AhR activation and the different methods used for measuring the autophagic flux. A previous study reported the proinflammatory effect of 6-formylindolo[3,2-b]carbazole (FICZ), an endogenous ligand for AhR, in vitro and the localized application in the EAE model [37], which is consistent with our results. Nevertheless, Di Meglio et al. reported that AhR activation by FICZ causes a reduction of inflammatory responses in a mouse model of psoriasis and human psoriatic skin [35]. This is in contrast with our study that showed how TCDD-mediated AhR activation induces inflammation in human keratinocytes and ex vivo skin biopsies. These opposite results could be explained as follows. First, conflicting data could be due to the different nature of the inflammatory skin response, which is provoked by a single agent and short-term exposure in skin biopsies or under in vitro conditions, whereas it is a multifactorial and chronic process in patients. Second, the effect of AhR might differ due to a heterogenous activation pattern in humans. Notably, the expression of a constitutively active form of AhR in keratinocytes causes pathologic skin lesions [38]. Prolonged AhR signaling in response to TCDD leads to its dysregulation [39]. On the contrary, physiological AhR activation in skin was found to be beneficial. Under physiological conditions, AhR signaling is supposed to be strictly controlled, and endogenous ligands such as FICZ are quickly metabolized by CYP enzyme, which is a downstream target of AhR, especially CYP1A1 [40].

In present study, the loss of AhR activity might impair p65 or p38 phosphorylation in an autophagy-independent manner as well as in an autophagy-dependent manner. AhR has various ligands and different actions according to its ligand. It also affects various complex responses following its stimulation or inhibition [41]. Canonical and non-canonical signaling pathways activated by AhR have been discovered [20]. In addition to well-known canonical AhR signaling, another pathway mediated by p65NF-kB among non-canonical AhR signaling has been identified [42]. Therefore, one single mechanism is not sufficient to explain the functions of AhR. Thus, more studies are required to unravel the functions and signaling pathways related to AhR.

This study has several limitations. First, rapamycin has no significant effects on LC3 expression in HaCaT cells, yet demonstrated significant induction of LC3 in NHEKs. These different effects of rapamycin on LC3 protein induction may have be caused by cell-specific effects. Moreover, in this study, the effect of autophagy induction by rapamycin was determined exclusively by monitoring changes in LC3 protein expression. Hence, to further verify the definite effect of rapamycin as an autophagy inducer it is important to also quantify the levels of other autophagy-related factors (ATG5, Beclin1, p62, etc.) via Western blotting, as well as by immunofluorescence assay and transmission electron microscopy to evaluate autophagosome formation. Second, we could not confirm the effect of AhR activation on autophagy nor the cross-talk between AhR and autophagy on skin inflammation in humans or animal models. Third, we could not evaluate the mechanisms by which AhR and autophagy regulate the adaptive immune system, for instance, by modulating the levels of cytokines critical in psoriasis pathogenesis, such as IL-17, IL-22, and IL-23 in the skin. Thus, further research is required to confirm our results in animal models and humans and to unravel the detailed mechanisms underlying the link between AhR and autophagy on the adaptive immune system.

In summary, our results demonstrate that AhR activation by environmental stimuli regulates autophagy, contributing to skin inflammation via the p65NF-κB/p38MAPK signaling pathways (Figure 10). Our data suggests that AhR-induced inhibition of autophagy may be involved in the pathogenesis of chronic inflammatory disorders such as psoriasis. Thus, AhR and autophagy modulation may serve as effective targets for therapeutic intervention in chronic inflammatory skin disease.

## 4. Materials and Methods

### 4.1. Patients and Sample Collections

Skin biopsy specimens were taken from patients with psoriasis (*n* = 6), as well as healthy volunteers (*n* = 6). All subjects provided written informed consent to participate in the study. The protocol was approved by the Institutional Review Board of Kangnam Sacred Heart Hospital (IRB no. 2018-05-024, 31th August 2018).

### 4.2. Cell Culture

The HaCaT (Human epidermal keratinocytes) cell line was procured from the National Centre for Cell Sciences (NCCS), Pune, India, and maintained in DMEM-F12 (Lonza, Walkersville, MD, USA) medium containing 10% (*v*/*v*) fetal bovine serum (FBS; Gibco, Thermo Fisher Scientific, Waltham, MA, USA), 100 units/mL penicillin, 100 mg/mL streptomycin, and 0.25 mg/mL amphotericin at 37 °C in a humidified atmosphere at 5% CO_2_. Then, 10 ng/mL TNF-α, IL-17A, IL-22, IL-1α, and Oncostatin-M (Prospec, East Brunswick, NJ) were included in the mixture of five proinflammatory cytokines (M5) [17]. HaCaT cells were stimulated with M5, causing inflammation and presenting a variety of features of psoriasis [17]. Normal human epidermal keratinocytes (NHEKs) were obtained as cryopreserved first-passage cells from Clonetics-Bio Whittaker (San Diego, CA, USA). They were grown in 100-mm tissue culture dishes in serum-free keratinocyte growth medium KGM-2 (Lonza, Walkersville, MD, USA) supplemented with bovine pituitary extract, human recombinant epidermal growth factor, insulin, hydrocortisone, transferrin, and epinephrine at 37 °C, 5% CO_2_. The media was replaced daily. When they reached 70–90% confluence, the cells were disaggregated with 0.25% trypsin/0.01% ethylenediaminetetraacetic acid (EDTA) in HEPES, and sub-cultured.

### 4.3. Ex Vivo Culture

The tissues were submerged in growth medium. A skin fragment was prepared, the fat tissue was removed, and the fragment was washed for 10 min in serum-free DMEM. The entire tissue was then placed on a multi-well plate (Costar, Cambridge, MA, USA) and cultivated for 48 h in 1 mL of DMEM at 37 °C, 5% CO_2_.

### 4.4. MTT Assay

Cell viability was determined by the conventional MTT reduction assay. Viable cells convert MTT to insoluble blue formazan crystals by mitochondrial respiratory chain enzyme succinate dehydrogenase. Cells were seeded at a density of 2 × 10^4^/well in a 6-well plate and maintained in 10% FBS-DMEM. Cells acquired a quiescent condition at confluence after incubation in serum-free DMEM for 24 h, followed by TCDD (Sigma-Aldrich, St. Louis, MO, USA), rapamycin (Sigma-Aldrich, St. Louis, MO, USA), and chloroquine (Sigma-Aldrich, St. Louis, MO, USA) treatment with the given concentrations of each compound for the indicated time. In treated cells, medium without the additives to be tested served as control. Cells were then washed with PBS and treated with MTT solution (final concentration, 0.5 mg/mL) for 4 h at 37 °C. The supernatant was removed and the formazan crystals were dissolved with 500 µL DMSO. Absorbance at 570 nm was measured with a microplate reader (Molecular Devices, Sunnyvale, CA, USA).

### 4.5. Immunohistochemistry

Immunohistochemistry was carried out in 10% formalin-fixed, paraffin-built tissues. The dissected tissues were washed several times with distilled water, and treated with 1% sodium borohydride for 1 h to remove any residual fixatives. The tissues were pretreated with 3% hydrogen peroxide solution for 10 min, washed with distilled water, and cultivated for 5 min with 1 TBST (Tris-buffered saline 0.1% Tween 20). To prevent non-specific reactions, the tissues were treated at room temperature around 20–22°C with normal goat serum (Vector Laboratories, Burlingame, CA, USA). Then, the tissues were cultivated overnight with rabbit anti-AhR (1:300; Abcam, Cambridge, MA, USA), rabbit anti-CYP1A1 (1:300; Abcam, Cambridge, MA, USA), and rabbit anti-LC3 (1:300; Novus Biologicals, Littleton, CO, USA). Tissues were washed with 1 TBST and incubated for 30 min at room temperature with the biotinylated secondary antibody solution from the Dako REAL Envision Detection System (Dako, Glostrup, Denmark). Next, tissues were washed with distilled water, counterstained with hematoxylin (Sigma-Aldrich, St. Louis, MO, USA), dehydrated and clarified using a conventional method, and prepared for Leica microsystems DFi8 LASX software light microscopy (Leica, Wetzlar, Germany). The level of staining was semi-quantitatively analyzed using LASX software (Leica, Wetzlar, Germany). The results are expressed as the mean optical density (± standard deviation) of six different digital images.

### 4.6. Western Blot Analyses

The cells were harvested in pro-prep lysis buffer (Intron, Seoul, Korea) with a protease inhibitor cocktail (Roche Diagnostics, Mannheim, Germany). The copper (II) sulfate solution in bicinchoninic acid solution (Sigma-Aldrich, St. Louis, MO, USA) was used to measure the protein concentrations. The same amount of protein (20 µg) was separated by 10% SDS-PAGE and transferred to enhanced chemiluminescence (ECL) nitrocellulose membranes (GE Healthcare, Buckinghamshire, UK), and then blocked for 1 h with 5% skim milk in TBST. The membranes were cultivated overnight at 4 °C with rabbit anti-AhR (1:1000, Abcam, Cambridge, MA), rabbit anti-CYP1A1 (1:1000, Abcam, Cambridge, MA, USA), rabbit anti-LC3 (1:1000, Abcam, Cambridge, MA, USA), rabbit anti-Beclin1 (1:1000, Novusbio, Centennial, CO, USA), rabbit anti-ATG5 (1:1000, Abcam, Cambridge, MA, USA), rabbit anti-P62 (1:1000, Abcam, Cambridge, MA, USA), rabbit anti-phosphor p65 (1:1000, Abcam, Cambridge, MA, USA), and rabbit anti-phosphor p38 (1:1000, Abcam, Cambridge, MA, USA) antibodies. The primary antibodies were detected with horseradish peroxidase-conjugated secondary antibodies (goat anti-rabbit, 1:1000; Abcam, Cambridge, MA, USA) and chemiluminescent luminol (LUMINOGRAPH II; Atto, Tokyo, Japan). Immunocomplexes were detected using an enhanced horseradish peroxidase/luminol chemiluminescence system (ECL Plus; Amersham International PLC, Little Chalfont, UK). Glyceraldehyde-3-phosphate dehydrogenase level was used as a loading control for Western blots.

### 4.7. Quantitative Reverse Transcriptase-PCR

Under the instructions of the manufacturer, the RNeasy Plus Mini Kit (Qiagen, Hilden, Germany) was used to extract total RNA. The transcriptor First Strand cDNA synthesis kit (Roche Applied Science, Mannheim, Germany) was used to synthesize cDNA from 1 µg of total RNA. Quantitative reverse transcriptase-PCR was carried out three times using the TaqMan master mix (Applied Biosystems, Foster City, CA, USA) and the Real-Time PCR System (Applied Biosystems). The primers used in this study for mRNA detection are as follows (Appendix A): *AHR* (TaqMan Assay ID Hs00169233_m1), *CYP1A1* (TaqMan Assay ID Hs1054794_m1), *LC3* (*MAP1LC3A*, TaqMan Assay ID Hs00176567_m1), *ATG5* (TaqMan Assay ID Hs00169468_m1), *BECN1* (TaqMan Assay ID Hs01007018_m1) *IL-1β* (TaqMan Assay ID Hs1555410_m1), *IL-6* (TaqMan Assay ID Hs00174131_m1), *TNF-α* (TaqMan Assay ID Hs00174128_m1) and *GAPDH* (TaqMan Assay ID 02758991_m1). The mRNA levels of *AHR, CYP1A1* and *LC3, ATG5*, *BECN1, IL-1β, IL-6*, and *TNF-α* were normalized to that of *GAPDH*. Relative quantification was performed using a Light Cycler^®^ 96 Instrument (Roche Diagnostics, Mannheim, Germany).

### 4.8. Transfection of siRNAs (Small Interfering RNA) Specific for AhR and ATG5

siRNAs targeting AhR (AhR siRNA: Ambion; Thermo Fisher Scientific, Waltham, MA, USA) or ATG5 (ATG5 siRNA: Ambion; Thermo Fisher Scientific, Waltham, MA, USA), as well as siRNA consisting of a scrambled sequence that would not lead to specific degradation of any cellular mRNA (control siRNA), were purchased from Ambion (Ambion; Thermo Fisher Scientific, Waltham, MA, USA). HaCaT cells cultured in 6-well plates were incubated for 48 h in 0.5 mL of culture medium with a mixture containing 5 nM siRNA and 3 µL of Lipofectamine RNAiMAX (Invitrogen; Thermo Fisher Scientific, Waltham, MA, USA). After 48 h cultivation, transfections were performed using the previously described short interfering RNA (siRNA) duplex directed against AhR or ATG5 mRNA sequence or control siRNA.

### 4.9. Statistical Analyses

Statistical analyses were conducted with GraphPad Prism version 5.01 (GraphPad Software, San Diego, CA, USA). Data were analyzed using Student’s *t*-test and one-way analysis of variance with Tukey’s post-test. All * *p*-values < 0.05 were considered statistically significant.

## Figures and Tables

**Figure 1 ijms-21-02195-f001:**
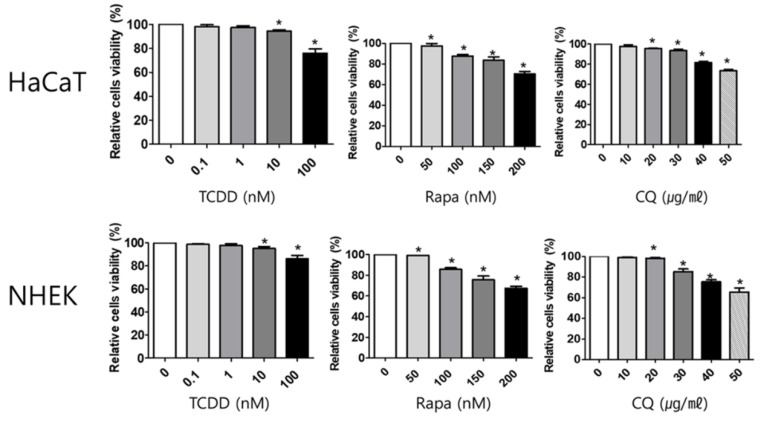
The effect of 2,3,7,8-Tetrachlorodibenzo-p-dioxin (TCDD), rapamycin, and chloroquine on HaCaT cells and NHEK viability. After treatment of HaCaT with (a) TCDD (0, 0.1, 1, 10, and 100 nM), (b) rapamycin (0, 50, 100, 150, and 200 nM), and (c) chloroquine (0, 10, 20, 30, 40, and 50 µg/mL) for 48 h, cell viability was measured by, 3-(4,5-dimethylthiazol-2yl)-2,5-diphenyl-tetrazolium bromide (MTT) assay. After treatment of NHEK with (d) TCDD (0, 0.1, 1, 10, and 100 nM), (e) rapamycin (0, 50, 100, 150, and 200 nM), and (f) chloroquine (0, 10, 20, 30, 40, and 50 µg/mL) for 48 h, cell viability was measured by MTT assay. Data represent the mean ± S.D. of three independent experiments (each performed in duplicate). Statistical significance was determined by one-way ANOVA followed multiple comparison Tukey’s test. * indicates *p* < 0.05 in comparison to unstimulated cells. Rapa, rapamycin, CQ, chloroquine, NHEK, normal human epidermal keratinocyte.

**Figure 2 ijms-21-02195-f002:**
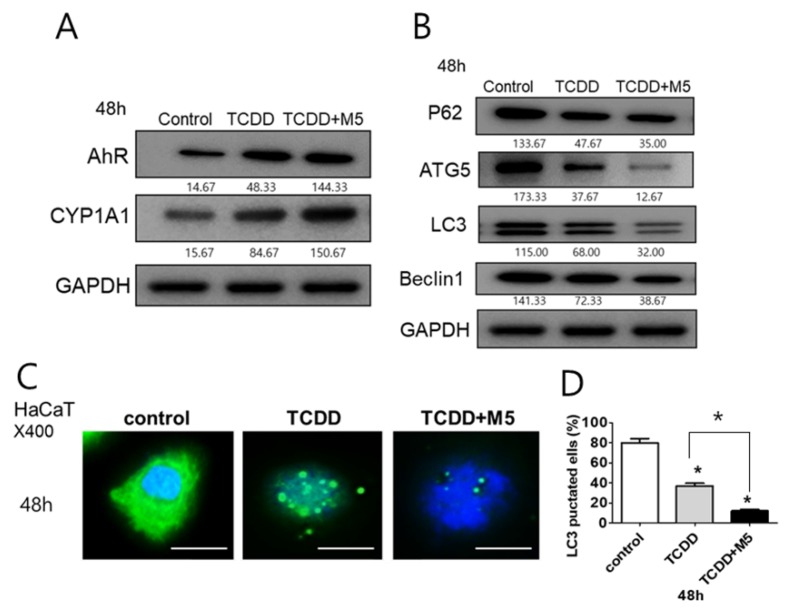
TCDD treatment in M5-stimulated HaCaT cells increased CYP1A1 and aryl hydrocarbon receptor (AhR) expression and decreased the expression of autophagy-related markers and autophagosome production. (**A**) Western blot analysis of HaCaT cells stimulated with TCDD (10 nM) and M5 (TNF-α, IL-17A, IL-22, IL-1α, and Oncostatin-M, each of 10 ng/mL) for 48 h using antibodies against AhR and CYP1A1; data normalization was based on Glyceraldehyde 3-phosphate dehydrogenase (GAPDH). The results are representative of three independent experiments. (**B**) Western blot analysis of HaCaT cells stimulated with TCDD (10 nM) and M5 (10 ng/ mL) for 48 h using antibodies against P62, ATG5, LC3, and Beclin-1; data normalization was based on GAPDH. The results are representative of three independent experiments. (**C,D**) Immunofluorescence analysis of LC3. Cells were stained with 4′,6-diamidino-2-phenylindole (DAPI) to visualize nuclei (blue), and immunolabeled with a combination of an anti-LC3 Ab and a fluorescein-5-isothiocyanate (FITC)-conjugated goat anti-rabbit IgG (green). FITC-conjugated goat anti-rabbit IgG heavy chain was used as the immunofluorescence staining marker. HaCaT cells and M5-stimulated HaCaT cells were treated with TCDD (10 nM) for 48 h. (**C**) Representative immunofluorescence images. Scale bar, 75 µm. (**D**) Percentages of cells with LC3-positive puncta. Data represent the mean ± S.D. of three independent experiments (each comprising at least 250 cells scored in five random fields.). Statistical significance was determined by one-way ANOVA followed multiple comparison Tukey’s test. * *p* < 0.05 in comparison to unstimulated cells.

**Figure 3 ijms-21-02195-f003:**
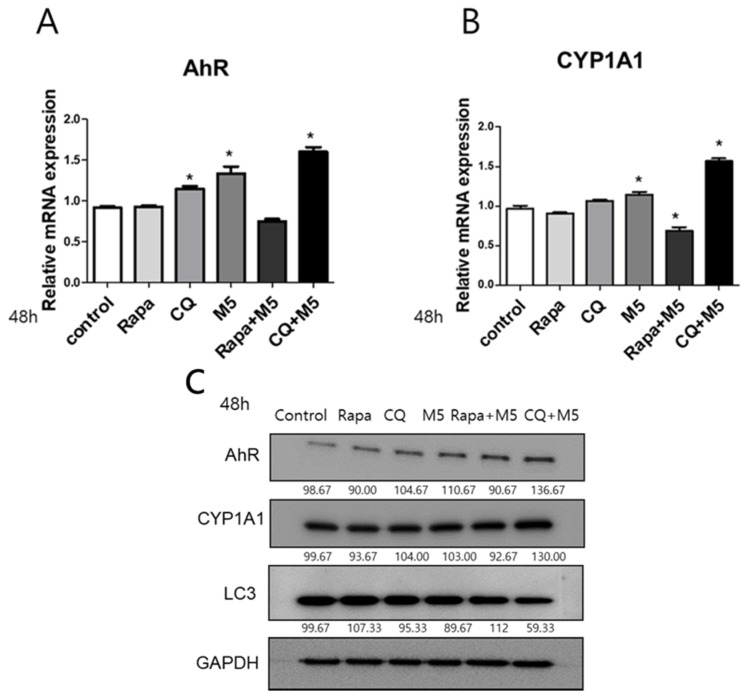
Autophagy inhibition significantly upregulated AhR, CYP1A1 genes, and proteins in M5-stimulated HaCaT cells. The mRNA was harvested from HaCaT cells treated with rapamycin (50 nM), CQ (20 µg/mL), M5 (10 ng/mL), rapamycin +M5, and CQ+M5 for 48 h. (**A**) AhR mRNA was detected by qPCR and relative expression was normalized to GAPDH. Data are presented as the mean ± S.D. of at least three independent experiments (each performed in duplicate). Statistical significance was determined by one-way ANOVA followed multiple comparison Tukey’s test. * *p* < 0.05 in comparison to unstimulated cells. (**B**) CYP1A1 mRNA was detected by qPCR and relative expression was normalized to GAPDH. Data are presented as the mean ± S.D. of at least three independent experiments (each performed in duplicate). Statistical significance was determined by one-way ANOVA followed multiple comparison Tukey’s test. * *p* < 0.05 in comparison to unstimulated cells. (**C**) Western blot analysis of HaCaT cells stimulated with 50 nM rapamycin, 20 µg/mL CQ, M5 (10 ng/mL), rapamycin +M5, or CQ+M5 for 48 h using antibodies against AhR and CYP1A1, and LC3; relative expression was normalized with GAPDH. The results are representative of three independent experiments. Rapa, rapamycin, CQ, chloroquine, GAPDH, glyceraldehyde 3-phosphate dehydrogenase.

**Figure 4 ijms-21-02195-f004:**
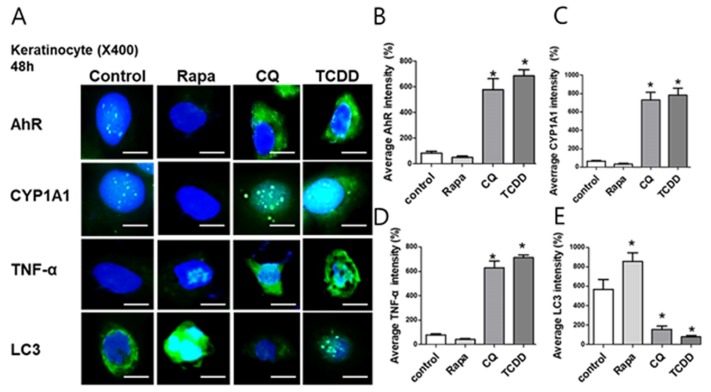
The effects of autophagy modulation and AhR activation on the expression of AhR, CYP1A1, TNF-α, and LC3. The level of AhR (**A,B**), CYP1A1 (**A,C**), TNF-α (**A,D**), and LC3 (**A,E**) expression was determined by immunofluorescence. Normal human epidermal keratinocytes (NHEKs) were treated with rapamycin (50 nM), CQ (20 µg/mL), and TCDD (10 nM) for 48 h. CYP1A1, TNF-α, and AhR expression was significantly increased in chloroquine- and TCDD-treated NHEKs. LC3 expression significantly decreased in chloroquine- and TCDD-treated NHEKs. Scale bar = 75 µm. Data represent the mean ± S.D. of three independent experiments (each performed in duplicate). Statistical significance was determined by one-way ANOVA followed multiple comparison Tukey’s test. * *p* < 0.05 in comparison to unstimulated cells. Rapa, rapamycin, CQ, chloroquine.

**Figure 5 ijms-21-02195-f005:**
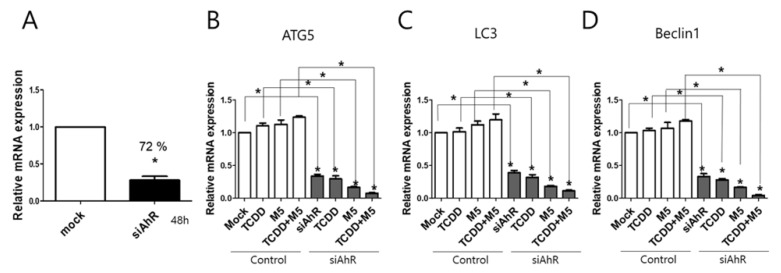
Knockdown of AhR resulted in significant attenuation of autophagy-related markers. (**A**) siRNA oligonucleotide targeting AhR was tested using qPCR. (**B–D**) HaCaT cells were transiently transfected with AhR siRNA and stimulated with TCDD (10 nM) in the presence of M5 (10 ng/mL) for 48 h. qPCR analysis was performed for (**B**) ATG5, (**C**) LC3, (**D**) Beclin1 expression. Data represent the mean ± S.D. of three independent experiments (each performed in duplicate). Statistical analysis was performed by one-way ANOVA test followed by multiple comparison Tukey’s test. * indicates *p* < 0.05 in comparison to unstimulated cells.

**Figure 6 ijms-21-02195-f006:**
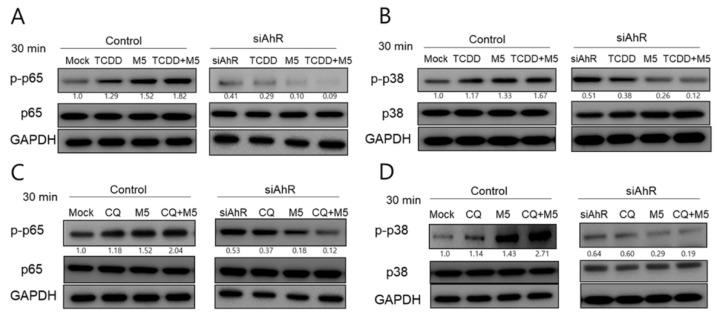
AhR siRNA transfection suppressed TCDD- and chloroquine-induced phosphorylation of p65NF-κB and p38MAPK in M5-stimulated HaCaT cells. (**A–D**) HaCaT cells were stimulated by M5 for 48 h before transfection. M5-stimulated HaCaT cells were transfected with AhR siRNA. At 48 h post-transfection, p65 and p38 phosphorylation was evaluated by Western blotting after a 30-min treatment with TCDD or chloroquine. The density of phosphorylated p38MAPK and p65NF-κB band was normalized to the corresponding loading control. Results are representative of three independent experiments. CQ, chloroquine.

**Figure 7 ijms-21-02195-f007:**
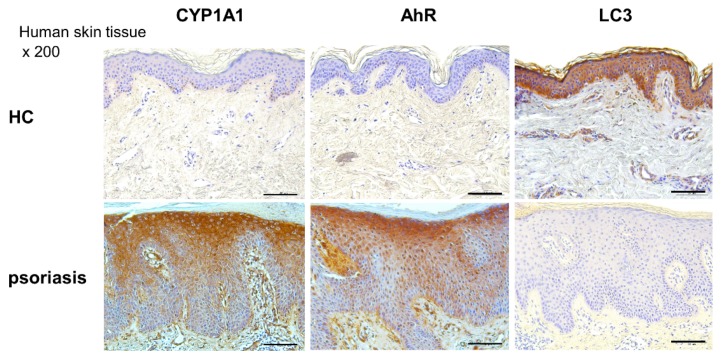
CYP1A1 and AhR expression was increased and LC3 decreased in psoriasis skin. Immunohistochemical staining of CYP1A1, AhR, and LC3 in human psoriasis lesion tissue and human healthy controls skin tissues. Bar = 50 µm. Data are representative of psoriasis (*n* = 3) and healthy volunteers (*n* = 3).

**Figure 8 ijms-21-02195-f008:**
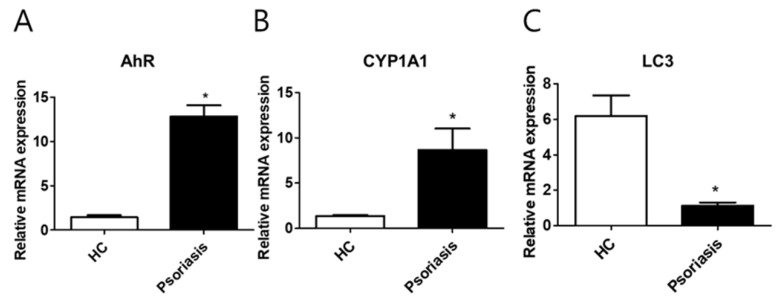
AhR, CYP1A1, and LC3 mRNA expression in psoriatic lesional skins. (**A–C**) qPCR analysis of CYP1A1, AhR, and LC3 expression in human psoriasis lesion (*n* = 3) and healthy control (HC) tissues (*n* = 3). Data represent the mean ± S.D. of three independent experiments (each performed in duplicate). Statistical significance was determined by Student’s *t*-test. * indicates *p* < 0.05 in comparison to unstimulated cells.

**Figure 9 ijms-21-02195-f009:**
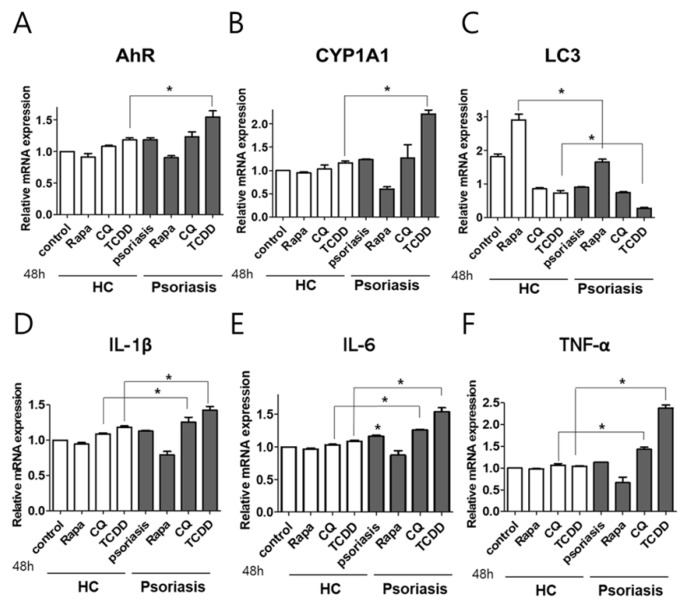
The effects of AhR or autophagy modulation on the expression of AhR, CYP1A1, LC3, and proinflammatory cytokines in human psoriasis skin biopsies. Human skin explants were treated with rapamycin, chloroquine, or TCDD, and cultured for 48 h before mRNA quantification. qPCR analysis was then performed to determine (**A**) AHR, (**B**) CYP1A1, (**C**) LC3, (**D**) IL-1β, (**E**) IL-6, and (**F**) TNF-α expression in human psoriasis skin biopsies (*n* = 3) and healthy control (HC) tissues (*n* = 3). Data represent the mean ± S.D. of three independent experiments (each performed in duplicate). Statistical analysis was performed by one-way ANOVA test followed by multiple comparison Tukey’s test. * indicates *p* < 0.05 in comparison to unstimulated cells. Rapa, rapamycin, CQ, chloroquine.

**Figure 10 ijms-21-02195-f010:**
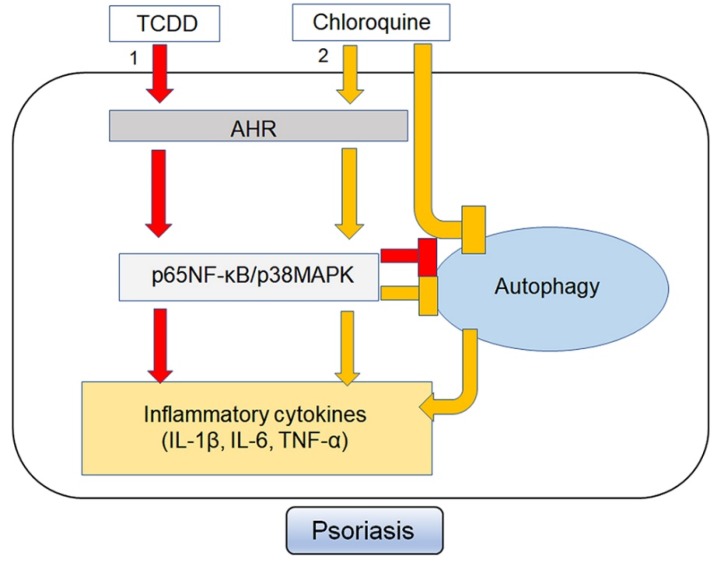
Suggested mechanism of AhR activation and autophagy on the pathogenesis of psoriasis. 1. AhR activation by TCDD suppress autophagy, contributing to skin inflammation via the p65NF-κB/p38MAPK signaling pathway. 2. Chloroquine, via inhibition of autophagy, leads to skin inflammation by AhR induction via the p65NF-κB/p38MAPK signaling pathway.

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
