# Peer review of "Role of Aryl Hydrocarbon Receptor Activation and Autophagy in Psoriasis-Related Inflammation"

_ijms, 2020, doi:10.3390/ijms21062195_

Round 1

Reviewer 1 Report

In the paper entitled "Role of Aryl Hydrocarbon Receptor Activation and

Autophagy in Psoriasis-related Inflammation" Kim and coworkers reported on the roles of AhR signaling and autophagy in the pathogenesis of chronic inflammatory skin disease, such as psoriasis. The authors hypothesize that AhR activation by TCDD could affect the autophagic

process contributing to the pathogenesis of chronic inflammatory skin disorder, including psoriasis.

They found that AhR-induced inhibition of autophagy has to be related to NF-κB/p65 and MAPK/p38 signaling pathway in keratinocytes.

These findings have been confirmed by experimental results reported in the manuscript.  

As stated by authors this paper does not clarify the exact mechanisms of cross-talk between AhR pathway and autophagy related pathways but lays the foundation for a series of experiment able to elucidating this relationships.

Only the clarification of these mechanisms will allow to employ AhR modulation for therapeutic treatment of skin chronic inflammatory diseases.  

The paper is well written and the data clearly presented.

The  introduction is in my opinion too short, the authors should better describe the role of AhR in autophagy and innate adaptive immunity because this is at the base of their research motivation and also because this of help for the readers.

Also the number of references relative to the above mentioned argument in introduction is not sufficient and they are old. For example you can cite Aryl Hydrocarbon Receptor Control of Adaptive Immunity, Pharmacol.  Rev. 2013 Oct; 65(4): 1148–1161.

Anyway I think that after the suggested modifications the paper can be published in IJMS.

Author Response

Response to Reviewer 1 Comments

In the paper entitled "Role of Aryl Hydrocarbon Receptor Activation and Autophagy in Psoriasis-related Inflammation" Kim and coworkers reported on the roles of AhR signaling and autophagy in the pathogenesis of chronic inflammatory skin disease, such as psoriasis. The authors hypothesize that AhR activation by TCDD could affect the autophagic process contributing to the pathogenesis of chronic inflammatory skin disorder, including psoriasis. They found that AhR-induced inhibition of autophagy has to be related to NF-κB/p65 and MAPK/p38 signaling pathway in keratinocytes. These findings have been confirmed by experimental results reported in the manuscript. As stated by authors this paper does not clarify the exact mechanisms of cross-talk between AhR pathway and autophagy related pathways but lays the foundation for a series of experiment able to elucidating this relationships. Only the clarification of these mechanisms will allow to employ AhR modulation for therapeutic treatment of skin chronic inflammatory diseases. The paper is well written and the data clearly presented.

Point 1. The introduction is in my opinion too short, the authors should better describe the role of AhR in autophagy and innate adaptive immunity because this is at the base of their research motivation and also because this of help for the readers.

Response 1. Thank you for this suggestion. Accordingly, we have provided a more detailed description on the role of AhR in autophagy and innate adaptive immunity within the revised Introduction as follows.:

“Recently, the roles of AhR and autophagy in innate and adaptive immunity have been demonstrated [17,18]. In fact, as a critical chemosensor, AhR activation regulates both innate and adaptive immune responses [19,20]. It was further identified that AhR affects the maturation and function of antigen-presenting on Langerhans cells [10]. AhR activation by TCDD and endogenous molecules also functions in the differentiation of T-helper 17 (Th17) and T regulatory (Treg) cells [21]. Additionally, autophagy also plays critical functions in modulating innate and adaptive immunity. In the innate immune response, autophagy not only contributes to the clearance of pathogens, but also participates in protection against toxins, and impacts cytokine production. In terms of the adaptive immunity, autophagy plays crucial roles in T cell maturation and differentiation, as well as antigen presentation [22]. However, the relationship between AhR and autophagy in the skin has not yet been elucidated. To the best of our knowledge, this study is the first to confirm the effects of AhR on autophagy and to describe a correlation between AhR and autophagy cross-talk and psoriasiform skin inflammation.” (line 66-79)

10.Jux, B.; Kadow, S.; Esser, C. Langerhans cell maturation and contact hypersensitivity are impaired in aryl hydrocarbon receptor-null mice. J. Immunol. 2009, 182, 6709-6717.

17.Xu, Y.; Jagannath, C.; Liu, X.D. Sharafkhaneh, A, Kolodziejska, K.E., Eissa, N.T. Toll-like receptor 4 is a sensor for autophagy associated with innate immunity. Immunity 2007, 27, 135-144.

18.Funatake, C.J., Marshall, N.B.; Steppan, L.B.; Mourich, D.V.; Kerkvliet, N.I. Cutting edge: activation of the aryl hydrocarbon receptor by 2,3,7,8-tetrachlorodibenzo-p-dioxin generates a population of CD4+ CD25+ cells with characteristics of regulatory T cells. J. Immunol. 2005, 175, 4184-4188.

19.Esser, C.; Rannug, A. The aryl hydrocarbon receptor in barrier organ physiology, immunology, and toxicology. Pharmacol Rev. 2015, 67, 259-279.

20.Quintana, F.J.; Sherr, D.H. Aryl hydrocarbon receptor control of adaptive immunity. Pharmacol Rev. 2013, 65, 1148-1161.

21.Quintana, F.J.; Basso, A.S.;Iglesias, A.H.; Korn, T.; Farez, M.F.; Bettelli, E.; Caccamo, M. Oukka, M.; Weiner, H.L. Control of T(reg) and T(H)17 cell differentiation by the aryl hydrocarbon receptor. Nature. 2008, 453, 65-71.

22.Cui, B.; Lin, H.; Yu, J.; Yu, J.; Hu, Z. Autophagy and the immune response. Adv Exp Med Biol. 2019, 1206, 595-634.

Point 2. Also the number of references relative to the above mentioned argument in introduction is not sufficient and they are old. For example you can cite Aryl Hydrocarbon Receptor Control of Adaptive Immunity, Pharmacol.  Rev. 2013 Oct; 65(4): 1148–1161.

Response 2. As per your suggestions we have cited additional references in the Introduction as follows:

13.Behrends, C.; Sowa, M.E.; Gygi, S.P.; Harper, J.W. Network organization of the human autophagy system. Nature. 2010, 466,68-76.

14.Zheng, H.Y.; Zhang, X.Y.; Wang, X.F.; Sun, B.C. Autophagy enhances the aggressiveness of human colorectal cancer cells and their ability to adapt to apoptotic stimulus. Cancer Biol Med. 2012, 9, 105-110

19.Esser, C.; Rannug, A. The aryl hydrocarbon receptor in barrier organ physiology, immunology, and toxicology. Pharmacol Rev. 2015, 67, 259-279.

20.Quintana, F.J.; Sherr, D.H. Aryl hydrocarbon receptor control of adaptive immunity. Pharmacol Rev. 2013, 65, 1148-1161.

21.Quintana, F.J.; Basso, A.S.;Iglesias, A.H.; Korn, T.; Farez, M.F.; Bettelli, E.; Caccamo, M. Oukka, M.; Weiner, H.L. Control of T(reg) and T(H)17 cell differentiation by the aryl hydrocarbon receptor. Nature. 2008, 453, 65-71.

22.Cui, B.; Lin, H.; Yu, J.; Yu, J.; Hu, Z. Autophagy and the immune response. Adv Exp Med Biol. 2019, 1206, 595-634.

Anyway I think that after the suggested modifications the paper can be published in IJMS.

  • We appreciate your insightful comments and taking the time to review our manuscript.

Reviewer 2 Report

The authors demonstrated that immune response of Aryl hydrocarbon receptor (AhR) and autophagy in the skin. For this study, they used TCDD ta activate AhR on autophagy in human keratinocytes. There’s lot of data which can support their theory.   But, this reviewer is kindly asking to check their manuscript with English and grammar. Also, some import information are missing. Although this manuscript is acceptable with sufficient amount of great data, I am strongly suggest to check their writing again vary carefully.

 1. In many parts, there is a need to check English grammar and convey clear meaning.For example, line 38, ‘TCDD is a known dioxin’ is needed to change ‘TCDD is known as dioxin.’ Also there are too many commas (,) in line 93 which  M5 is though to be an individually named cytokine instead of general M5. It is same point of comment No 2 in below. 

2. Generally, there are too many commas (,) in a sentence, so what you're trying to say isn't exactly clear. For example, line 43 “TCDD has multiple biological and pathologic effects in humans, even at low doses” is suggested to change ‘TCDD has multiple biological and pathological effects on humans even at low doses. Therefore, it is required to use more accurate English for the whole story flow and to avoid too many comma symbols. 

3. Authors used different types of P value based on significance. In general, p value less than 0.05 is considered to be a difference, so I  recommend to authors not to divide so detailed (0.01, 0.001)  but to unite less than p<0.05. 

4. Reviewer suggest to authors to describe or comment that “LC3 expression was reduced under Rapa+M5 conditions but it is not significantly different” in Fig 3F.  

5. Exactly same sentences are repeated at 419-421 and 432-435. Please change the 432-435 to ‘The level of staining was analyzed with same method of immunohistology which described in above’  

6. Authors analyzed mRNA expression but still reviewer can not see the information of each primer. Please provide the primer sequences as a table if necessary which were used for qPCR. It is important for next investigators.

Author Response

Response to Reviewer 2 Comments

The authors demonstrated that immune response of Aryl hydrocarbon receptor (AhR) and autophagy in the skin. For this study, they used TCDD ta activate AhR on autophagy in human keratinocytes. There’s lot of data which can support their theory.   But, this reviewer is kindly asking to check their manuscript with English and grammar. Also, some import information are missing. Although this manuscript is acceptable with sufficient amount of great data, I am strongly suggest to check their writing again vary carefully.

We appreciate your insightful comments. Please find our point-by-point responses to each comment. We thank you for your consideration and look forward to your response.

Point 1. In many parts, there is a need to check English grammar and convey clear meaning. For example, line 38, ‘TCDD is a known dioxin’ is needed to change ‘TCDD is known as dioxin.’ Also there are too many commas (,) in line 93 which M5 is thought to be an individually named cytokine instead of general M5. It is same point of comment No 2 in below. 

Response 1. Thank you for these comments. Accordingly, we have had the manuscript revised for English and grammar by a native English-speaking. We have also corrected the specific sentences indicated by the reviewer.

(line 36, TCDD is a known dioxin -->TCDD is known as dioxin

line 118-119, proinflammatory cytokines, namely M5, TNF-α, IL-17A, IL-22, IL-1α, and Oncostatin-M that --> proinflammatory cytokines, namely M5, that)

Point 2. Generally, there are too many commas (,) in a sentence, so what you're trying to say isn't exactly clear. For example, line 43 “TCDD has multiple biological and pathologic effects in humans, even at low doses” is suggested to change ‘TCDD has multiple biological and pathological effects on humans even at low doses. Therefore, it is required to use more accurate English for the whole story flow and to avoid too many comma symbols. 

Response 2. Thank you for these suggestions. We have revised the manuscript for improved punctuation to ensure better overall flow and readability of the manuscript.:

Line 41-42, specifically, has been changed from “TCDD has multiple biological and pathologic effects in humans, even at low doses” --> “TCDD has multiple biological and pathological effects on humans even at low doses.”.

Point 3. Authors used different types of P value based on significance. In general, p value less than 0.05 is considered to be a difference, so I recommend to authors not to divide so detailed (0.01, 0.001) but to unite less than p<0.05. 

Response 3. As per the reviewer’s comments we have united the p values (less than p < 0.05) throughout the manuscript.

Point 4. Reviewer suggest to authors to describe or comment that “LC3 expression was reduced under Rapa+M5 conditions but it is not significantly different” in Fig 3F.  

Response 4. We appreciate your helpful comment. Unfortunately, these descriptions could not be added because other reviewer recommended that bar graphs including Fig. 3F should be removed.

Point 5. Exactly same sentences are repeated at 419-421 and 432-435. Please change the 432-435 to ‘The level of staining was analyzed with same method of immunohistology which described in above’  

Response 5. Thank you for your helpful comment. However, these descriptions could not be included because the results of immunofluorescence staining have been deleted for better overall flow of the manuscript.

Point 6. Authors analyzed mRNA expression but still reviewer can not see the information of each primer. Please provide the primer sequences as a table if necessary which were used for qPCR. It is important for next investigators.

Response 6. We apologize for this oversight. We have since added all primer information in Supplementary Table 2 and have referenced this in the Materials and Methods section of the revised manuscript.

“The primers used in this study for mRNA detection are as follows: AHR (TaqMan Assay ID Hs00169233_m1), CYP1A1 (TaqMan Assay ID Hs1054794_m1), LC3 (MAP1LC3A, TaqMan Assay ID Hs00176567_m1), ATG5 (TaqMan Assay ID Hs00169468_m1), BECN1 (TaqMan Assay ID Hs01007018_m1), IL-1β (TaqMan Assay ID Hs1555410_m1), IL-6 (TaqMan Assay ID Hs00174131_m1), TNF-α (TaqMan Assay ID Hs00174128_m1), and GAPDH (TaqMan Assay ID 02758991_m1).” (line,570-574)

Supplementary Table 2.the used mRNA primers

Genes

Product numbers

AHR

TaqMan Assay ID Hs00169233_m1

CYP1A1

TaqMan Assay ID Hs1054794_m1

LC3

TaqMan Assay ID Hs00176567_m1

ATG5

TaqMan Assay ID Hs00169468_m1

BECN1

TaqMan Assay ID Hs01007018_m1

IL-1β

TaqMan Assay ID Hs1555410_m1

IL-6

TaqMan Assay ID Hs00174131_m1

TNF-α

TaqMan Assay ID Hs00174128_m1

GAPDH

TaqMan Assay ID 02758991_m1

Reviewer 3 Report

Dear Authors,

Results presented in this manuscript are interesting, especially since the changes observed in the mRNA have been confirmed at the protein level. However, we recommend reducing the size of the legends to make the article easier to read. It should be avoided to describe the results in the figure legend and simply mention the details necessary for understanding.

Comments:

Introduction: The introduction is a little hasty. Please mention what is LC3 and how the effect of AhR on autophagy can influence psoriasis. Since you don’t use the IMQ-induced psoriasis model, there is no need to describe it.

Line 82-83: You said that the chosen concentrations were not cytotoxic. However, in your graph there is a significant decrease in comparison to unstimulated cells. Can you really tell that it is non cytotoxic and why did you choose these concentrations anyway? Please, rephrase if so and tell that it is in an acceptable range.

Figure 1: Reduce the y axis to 100.

Figure 1: Please uniform the p-value section. * indicates p value < 0.05 and ** p value < 0.01 and *** p value < 0.001

Line 94: Can you add Supplementary Materials showing those features of psoriasis.

Line 97: (Fig. 2b-h) Is it rather (Fig. 2d-h)?

Figure 2E: The difference is difficult to see between the TCDD group and the TCDDD+M5 group. Would it be possible to assess whether this difference is significant?

Figure 2: The results are already described in the results section. Since the paper is already long, this repetitive information only lengthens the legend. Please delete these lines:

  1. Lines 107-108: TCDD significantly upregulated CYP1A1 and AhR protein expression in M5-stimulated HaCaT cells.
  2. Lines 111-113: The expression of autophagy-related proteins was strongly decreased in both TCDD- and M5-treated HaCaT cells.
  3. Lines 116-117: The production of autophagosomes was particularly suppressed M5-stimulated HaCaT cells with TCDD treatment.

Figure 2I: Please mention what is the marker used for the immunofluorescence (and also in the legend).

Figure 2I: The scale bar is incorrect and not clear. Please confirm that it is 75µm (?) or 400µm (?) as in the legend although it seems too high for only one keratinocyte.

Figure 3: Please shorten the legend by removing the results described elsewhere (lines 137-138, 144-145, …). Moreover, since the M5 conditions is already describe, it is not necessary to detail it (lines 134 and 142).

Line 147: Why did you change the cell type (HaCaT to NHEKs)?

Figure 4: There is no scale bar for the immunofluorescence staining. Please add one and confirm that it is indeed 400µm as mentioned in the legend, although that seems a little high.

Figure 4: line 161 “Rapa: rapamycin; CQ: chloroquine”. Moreover, this description of the abbreviation does not appear in almost any figure. Please add it in Figures 1, 2, 3, 5, 7, 8, 9, 10

Line 162: Please revise the structure of the following sentence: “AhR Knockdown Significantly Attenuated Rapamycin–induced Downregulation of AhR Expression and ATG5 Knockdown Decreased TCDD-induced Suppression of Autophagy-related Marker Expression

Lines 169-170: Please revise the structure of the following sentence: “AhR knockdown resulted in a significant attenuation of rapamycin induced suppression of AhR and CYP1A1 expression in HaCaT cells.”

Line 175-177: Please revise the structure of the following sentence: “siRNA-induced ATG5 knockdown resulted in a significant attenuation of TCDD-induced decrease in expression of autophagy-related factors, …”

Figure 5: Same as in figure 4 (there is no scale bar in the figure but there is one mentioned in the legend. Please remove all result interpretations of the legend (increases, decreases, etc). Also indicate in the legend what the crosses (†) observed in the figure represent.

Figure 5H: Please mention what is the marker used for the immunofluorescence (and also in the legend).

Figure 7: Please indicate in the legend what the crosses (†) observed in the figure represent.

Figure 7: Please add the housekeeping protein for Western blot analysis.

Lines 219-220: “Taken together, our results showed a critical role of AhR in both TCDD- and chloroquine-induced p38MAPK/p65NF-κB phosphorylation.” This sentence should appear in the discussion and not in the result section.

Figure 8: The use of “non-psoriasis” in the figure can be confusing. Rather indicate non-lesional (psoriasis) and lesional (psoriasis).

Figure 8: The results with the AD lesional skin are not necessary in the study. Please remove it or discuss more about it in the discussion section while moving them in the supplementary material.

Figure 8: It is not clear whether there is or not a scale bar in the immunohistochemistry pictures.

Figure 8E: There is no scale bar information in the legend and the scale bar in the figure is not clear. Is it really 400X?!

Figure 10: Please indicate in the legend what the crosses (†) observed in the figure represent.

Line 295: Can you give more details.

Lines 292-303: Lack of consistency on this section of the discussion.

Line 306: Please add references.

Line 306: Please rephrase this sentence: “Thus, our study suggested hypothesized that AhR activation by environmental pollutants such as dioxin (TCDD)…”

Line 378: Why choose 10ng/ml for each cytokine in the mixture?

Line 376: “… fetal bovine serum (FBS; Gibco, Thermo Fisher Scientific, USA), …”

Lines 378, 385, 392: Please change CO2 for CO2.

Line 396: please change “… density of 2×104/6-well plates…” for “… density of 2×104/well in a 6-well plate…”

Line 401: Does the test was optimized before because the final concentration of MTT solution is a little high and this can explain why you observed a decrease in the cell viability. Usually, a concentration of 0.5mg/ml (for 1 to 4 hours) is used in those assays.

Line 486: Please update this list with all the abbreviations (Rapa, CQ, FBS, FICZ, …, are missing).

Figures in general: There is no need to use two types of identification for every condition. For an example (figure 2a) you use Control, TCDD, TCDD+M5 and you also use it with + and – on top of the first identification method. It is a little repetitive. 

Author Response

Response to Reviewer 3 Comments

Dear Authors,

Results presented in this manuscript are interesting, especially since the changes observed in the mRNA have been confirmed at the protein level. However, we recommend reducing the size of the legends to make the article easier to read. It should be avoided to describe the results in the figure legend and simply mention the details necessary for understanding.

We appreciate the insightful comments on our manuscript. Please find our point-by-point responses to each comment. As per your suggestion, we have reduced the size of the legends to make the article easier to read. We hope that the manuscript is now suitable for publication in the International Journal of Molecular Sciences. We thank you for your consideration and look forward to your response.

Comments:

Point 1. Introduction: The introduction is a little hasty. Please mention what is LC3 and how the effect of AhR on autophagy can influence psoriasis. Since you don’t use the IMQ-induced psoriasis model, there is no need to describe it.

Response 1. Thank you for this comment. As per your suggestion we have expanded our original description on LC3 as follows:

Lines 55-59: “Autophagy is controlled by the Atg (autophagy-related) family of proteins, among which the ubiquitin- like protein, microtubule-associated protein IA/IB light chain 3 (LC3), is involved in cytosolic cargo recruitment and autophagosome formation [13]. Hence, it is commonly used as an autophagy marker to indicate autophagosome quantity as the lipidated form of LC3 accumulates in autophagosomal membranes [14].”

  1. Behrends, C.; Sowa, M.E.; Gygi, S.P.; Harper, J.W. Network organization of the human autophagy system. Nature. 2010, 466,68-76.
  2. Zheng, H.Y.; Zhang, X.Y.; Wang, X.F.; Sun, B.C. Autophagy enhances the aggressiveness of human colorectal cancer cells and their ability to adapt to apoptotic stimulus. Cancer Biol Med. 2012, 9, 105-110

We have also provided additional information on the effect of AhR on autophagy and their role in psoriasis as follows:

Lines 66-79: “Recently, the roles of AhR and autophagy in innate and adaptive immunity have been demonstrated [17,18]. In fact, as a critical chemosensor, AhR activation regulates both innate and adaptive immune responses [19,20]. It was further identified that AhR affects the maturation and function of antigen-presenting on Langerhans cells [10]. AhR activation by TCDD and endogenous molecules also functions in the differentiation of T-helper 17 (Th17) and T regulatory (Treg) cells [21]. Additionally, autophagy also plays critical functions in modulating innate and adaptive immunity. In the innate immune response, autophagy not only contributes to the clearance of pathogens, but also participates in protection against toxins, and impacts cytokine production. In terms of the adaptive immunity, autophagy plays crucial roles in T cell maturation and differentiation, as well as antigen presentation [22]. However, the relationship between AhR and autophagy in the skin has not yet been elucidated. To the best of our knowledge, this study is the first to confirm the effects of AhR on autophagy and to describe a correlation between AhR and autophagy cross-talk and psoriasiform skin inflammation.”

  • Jux, B.; Kadow, S.; Esser, C. Langerhans cell maturation and contact hypersensitivity are impaired in aryl hydrocarbon receptor-null mice. J. Immunol. 2009, 182, 6709-6717.
  • Xu, Y.; Jagannath, C.; Liu, X.D. Sharafkhaneh, A, Kolodziejska, K.E., Eissa, N.T. Toll-like receptor 4 is a sensor for autophagy associated with innate immunity. Immunity 2007, 27, 135-144.
  • Funatake, C.J., Marshall, N.B.; Steppan, L.B.; Mourich, D.V.; Kerkvliet, N.I. Cutting edge: activation of the aryl hydrocarbon receptor by 2,3,7,8-tetrachlorodibenzo-p-dioxin generates a population of CD4+ CD25+ cells with characteristics of regulatory T cells. J. Immunol. 2005, 175, 4184-4188.
  • Esser, C.; Rannug, A. The aryl hydrocarbon receptor in barrier organ physiology, immunology, and toxicology. Pharmacol Rev. 2015, 67, 259-279.
  • Quintana, F.J.; Sherr, D.H. Aryl hydrocarbon receptor control of adaptive immunity. Pharmacol Rev. 2013, 65, 1148-1161.
  • Quintana, F.J.; Basso, A.S.;Iglesias, A.H.; Korn, T.; Farez, M.F.; Bettelli, E.; Caccamo, M. Oukka, M.; Weiner, H.L. Control of T(reg) and T(H)17 cell differentiation by the aryl hydrocarbon receptor. Nature. 2008, 453, 65-71.
  • Cui, B.; Lin, H.; Yu, J.; Yu, J.; Hu, Z. Autophagy and the immune response. Adv Exp Med Biol. 2019, 1206, 595-634.

Finally, in the Introduction we have deleted the description on the IMQ-induced psoriasis model.

Point 2. Line 82-83: You said that the chosen concentrations were not cytotoxic. However, in your graph there is a significant decrease in comparison to unstimulated cells. Can you really tell that it is non cytotoxic and why did you choose these concentrations anyway? Please, rephrase if so and tell that it is in an acceptable range.

Response 2. As you have pointed out, in Fig. 1, there are significant decreases in viability compared to unstimulated cell. However, the cell viability in the chosen concentrations were all above 90%. (line 98-101) Generally, the cell viability over 90% is considered to be an acceptable range. Further, previous studies using TCDD, rapamycin and chloroquine, also employed the same concentrations as those used in our study.

TCDD

1)Aryl Hydrocarbon Receptor Modulates the Expression of TNF-α and IL-8 in Human Sebocytes via the MyD88-p65NF-κB/p38MAPK Signaling Pathways. J Innate Immun. 2019;11(1):41-51.

2) 2,3,7,8-Tetrachlorodibenzo-p-dioxin Increases the Expression of Genes in the Human Epidermal Differentiation Complex and Accelerates Epidermal Barrier Formation. Toxicol Sci. 2011 Nov;124(1):128-37

Chloroquine

1) Chloroquine treatment of ARPE-19 cells leads to lysosome dilation and intracellular lipid accumulation: Possible implications of lysosomal dysfunction in macular degeneration. Cell Biosci. 2011 Mar 8;1(1):10.

2) Quercetin induces protective autophagy in gastric cancer cells: Involvement of Akt-mTOR- and hypoxia-induced factor 1α-mediated signaling. Autophagy. 2011 Sep;7(9):966-78.

Rapamycin

Synergistic Inhibition of Tumor Necrosis Factor-Alpha-Stimulated Pro-Inflammatory Cytokine Expression in HaCaT Cells by a Combination of Rapamycin and Mycophenolic Acid. Ann Dermatol. 2015 Feb;27(1):32-9.

Point 3. Figure 1: Reduce the y axis to 100.

Response 3. Thank you for your suggestion. We have reduced the y-axis to 100 in Fig. 1.

Point 4. Figure 1: Please uniform the p-value section. * indicates p value < 0.05 and ** p value < 0.01 and *** p value < 0.001

Response 4. Thank you for this suggestion. Accordingly, we have uniformed the p-value section as * indicates p value < 0.05 according to the suggestion of another reviewer to unite all p values as p < 0.05.

Point 5. Line 94: Can you add Supplementary Materials showing those features of psoriasis.

Response 5. We have added Supplementary Materials showing the characteristics of M5 on features of psoriasis.

Supplementary Table 1. The effects of M5 on chemokines and antimicrobial peptides production [16].

Keratinocyte cultures

Human skin explants

Animal model

CXCL chemokines production

CXCL1, CXCL5, CXCL8 ↑

CXCL8 ↑

CXCL1, CXCL2, CXCL3 ↑

Chemotactic activity

Neutrophil chemotactic activity↑

Antimicrobial peptides production and activity

BD2, BD3, S100A7 ↑

Antibacterial activity to E.coli

BD2, S100A7 ↑

BD2, S100A7 ↑

Transcriptional profile change

BD2, BD3, LL37, RNASE7, PI3, S100A7, S100A7A, S100A12, CXCL1, CXCL2, CXCL3, CXCL5, CXCL6, CXCL8 ↑

  • Guilloteau, K.; Paris, I.; Pedretti, N.; Boniface, K.; Juchaux, F.; Huguier, V.; Guillet, G.; Bernard, F.X.; Lecron, J.C.; Morel, F. Skin Inflammation Induced by the Synergistic Action of IL-17A, IL-22, Oncostatin M, IL-1α, and TNF-α Recapitulates Some Features of Psoriasis. J Immunol. 2010, 184, 5263-5270.

Point 6. Line 97: (Fig. 2b-h) Is it rather (Fig. 2d-h)?

Response 6. We have corrected in line 123 (Fig. 2b-h) à (Fig. 2b) since we have deleted the western blot analysis bar graphs (previous Fig. 2B,C,E,F,G,H) and added the individual values of densitometrical analysis in the western blot images instead.

Point 7. Figure 2E: The difference is difficult to see between the TCDD group and the TCDD+M5 group. Would it be possible to assess whether this difference is significant?

Response 7. As per your suggestion we have assessed whether the difference is significant between the TCDD group and the TCDD+M5 group in the previous Fig. 2E. We found a significant difference between these groups. However, these descriptions could not be added because other reviewer recommended that bar graphs should be removed. Further, we also confirmed that the difference was significant between the TCDD group and the TCDD+M5 group in autophagosome formation. This description was added in revised Fig. 2D and result section (line 126-127).

Point 8. Figure 2: The results are already described in the results section. Since the paper is already long, this repetitive information only lengthens the legend. Please delete these lines:Lines 107-108: TCDD significantly upregulated CYP1A1 and AhR protein expression in M5-stimulated HaCaT cells.

Lines 111-113: The expression of autophagy-related proteins was strongly decreased in both TCDD- and M5-treated HaCaT cells.

Lines 116-117: The production of autophagosomes was particularly suppressed M5-stimulated HaCaT cells with TCDD treatment.

Response 8. Thank you for your comments. We have deleted all repetitive descriptions in Fig. 2 legends.

Point 9. Figure 2I: Please mention what is the marker used for the immunofluorescence (and also in the legend).

Response 9. We apologize for this oversight. We used FITC as the marker of immunofluorescence and have added the descriptions in the Fig. 2 legend. “FITC-conjugated goat anti-rabbit IgG heavy chain was used as the IF marker.”

Point 10. Figure 2I: The scale bar is incorrect and not clear. Please confirm that it is 75µm (?) or 400µm (?) as in the legend although it seems too high for only one keratinocyte.

Response 10. In previous manuscript, we presented both figures with original magnification x 200 and the figure with original magnification x 400. However, we understand how this may have been confusing. We apologize for this oversight. We have, therefore, removed the figures with original magnification x 200 and have presented the figures with a scale bar (75 µm).

Point 11. Figure 3: Please shorten the legend by removing the results described elsewhere (lines 137-138, 144-145, …). Moreover, since the M5 conditions is already describe, it is not necessary to detail it (lines 134 and 142).

Response 11. As per your suggestion, we have shortened the legends of Fig.3 by removing repetitive descriptions.

Point 12. Line 147: Why did you change the cell type (HaCaT to NHEKs)?

Response 12. To further confirm the effects of TCDD, Rapa, and CQ on the expression of AhR and autophagy-related factors, we performed immunofluorescence analysis in NHEKs. The experiments with other cell lines often show different results under the same treatments. Therefore, we also performed the experiments with other cell lines (NHEKs) to definitively demonstrate our results.

Point. 13. Figure 4: There is no scale bar for the immunofluorescence staining. Please add one and confirm that it is indeed 400µm as mentioned in the legend, although that seems a little high.

Response 13. We apologize for this oversight. We have added scale bars to Fig. 4 and have confirmed that it was 75 µm.

Point 14. Figure 4: line 161 “Rapa: rapamycin; CQ: chloroquine”. Moreover, this description of the abbreviation does not appear in almost any figure. Please add it in Figures 1, 2, 3, 5, 7, 8, 9, 10

Response 14. We have added abbreviations for Rapa and CQ in the legends for Fig. 1, 2, 3, 5, 7, 8, 9, and 10.

Point 15. Line 162: Please revise the structure of the following sentence: “AhR Knockdown Significantly Attenuated Rapamycin–induced Downregulation of AhR Expression and ATG5 Knockdown Decreased TCDD-induced Suppression of Autophagy-related Marker Expression

Response 15. We have revised the sentence as “AhR Knockdown Significantly Attenuated TCDD–induced Suppression of Autophagy “. (line 216-218)

Point 16. Lines 169-170: Please revise the structure of the following sentence: “AhR knockdown resulted in a significant attenuation of rapamycin induced suppression of AhR and CYP1A1 expression in HaCaT cells.”

Response 16. Thank you for your comment. However, we have removed this sentence according to the suggestion of another reviewer.

Point 17. Line 175-177: Please revise the structure of the following sentence: “siRNA-induced ATG5 knockdown resulted in a significant attenuation of TCDD-induced decrease in expression of autophagy-related factors, …”

Response 17. Thank you for helpful comment. We have deleted this sentence for more accurate understanding as per another reviewer’s suggestion.

Point 18. Figure 5: Same as in figure 4 (there is no scale bar in the figure but there is one mentioned in the legend. Please remove all result interpretations of the legend (increases, decreases, etc). Also indicate in the legend what the crosses (†) observed in the figure represent.

Response 18. We apologize for this oversight. We have since added scale bars in Fig. 5. We also have removed all result interpretations from Fig. 3 legends (increases, decreases, etc). Lastly, all crosses (†) were replaced by *.

Point 19. Figure 5H: Please mention what is the marker used for the immunofluorescence (and also in the legend).

Response 19. We used FITC as the marker of immunofluorescence. We have added the descriptions in the Fig. 5H legend. “FITC-conjugated goat anti-rabbit IgG heavy chain was used as the IF marker.”

Point 20. Figure 7: Please indicate in the legend what the crosses (†) observed in the figure represent.

Response 20. We have replaced the crosses (†) with * in all figures and have provided corresponding descriptions in the legends (of Fig. 6 in revised manuscript).

Point 21. Figure 7: Please add the housekeeping protein for Western blot analysis.

Response 21. We appreciate your helpful comment. In the western blot analysis of Fig. 7 (Fig 6 in revised manuscript), we used GAPDH as the housekeeping protein. We apologize for not including this in the original manuscript results. We have revised Fig. 7 (Fig 6 in revised manuscript) to include reference to GAPDH.

Point 22. Lines 219-220: “Taken together, our results showed a critical role of AhR in both TCDD- and chloroquine-induced p38MAPK/p65NF-κB phosphorylation.” This sentence should appear in the discussion and not in the result section.

Response 22. As per your suggestion, we have removed the sentences from lines 292-293.

Point 23. Figure 8: The use of “non-psoriasis” in the figure can be confusing. Rather indicate non-lesional (psoriasis) and lesional (psoriasis).

Response 23. Thank you for helpful comment. However, we have deleted this figure for more accurate understanding as per another reviewer’s suggestion.

Point 24. Figure 8: The results with the AD lesional skin are not necessary in the study. Please remove it or discuss more about it in the discussion section while moving them in the supplementary material.

Response 24. As per your recommendation, we have removed the results with the AD lesional skin (Fig 7 in revised manuscript).

Point 25. Figure 8: It is not clear whether there is or not a scale bar in the immunohistochemistry pictures.

Response 25. We have added scale bars for the IHC images in Figure 8 (Fig 7 in revised manuscript).

Point 26. Figure 8E: There is no scale bar information in the legend and the scale bar in the figure is not clear. Is it really 400X?!

Response 26. We apologize for this oversight. We have confirmed that the scale was 50 µm. However, we have deleted this figure for more accurate understanding as per another reviewer’s suggestion.

Point 27. Figure 10: Please indicate in the legend what the crosses (†) observed in the figure represent.

Response 27. We have replaced the crosses (†) with * in all figures.

Point 28. Line 295: Can you give more details.

Response 28. As per your request, we have expanded on the description on line 295 as follows:

Lines 391-399: “A recent study reported that single-nucleotide polymorphisms (SNPs) in the ATG16L1 gene (rs10210302, rs12994971, rs2241880, rs2241879, and rs13005285) are linked to psoriasis susceptibility [30]. The ATG16L1 protein, a major component of the autophagy-related protein complex, is critical to the autophagy process [31]. Moreover, it has been speculated that an ATG16L1 defect affects the role of autophagy machinery in various signaling pathways involved in modulating cytokine production, resulting in the accumulation of dysfunctional proteins and organelles, and subsequent chronic inflammation [31].”

  • Douroudis, K.; Kingo, K.; Traks, T., Reimann, E.; Raud, K.; Ratsep, R.; Mössner R.; Silm H.; Vasar E.; Kõks S.Polymorphisms in the ATG16L1 gene are associated with psoriasis vulgaris. Acta Derm. Venereol. 2012, 92, 85-87.
  • Mizushima, N.; Kuma, A.; Kobayashi, Y.; Yamamoto, A.; Matsubae, M.; Takao, T.; Natsume, T.; Ohsumi, Y.; Yoshimori, T. Mouse Apg16L, a novel WD-repeat protein, targets to the autophagic isolation membrane with the Apg12-Apg5 conjugate. J Cell Sci. 2003, 116, 1679–1688.

Point 29. Lines 292-303: Lack of consistency on this section of the discussion.

Response 31. This paragraph was revised to ensure consistency of topic.

Lines 388-403: “Autophagy has been regarded as an endogenous defense mechanism against environmental disturbances. Notably, in addition to maintaining skin homeostasis, autophagy has also been implicated in the development of skin disorders [29]. A recent study reported that single-nucleotide polymorphisms (SNPs) in the ATG16L1 gene (rs10210302, rs12994971, rs2241880, rs2241879, and rs13005285) are linked to psoriasis susceptibility [30]. The ATG16L1 protein, a major component of the autophagy-related protein complex, is critical to the autophagy process [31]. Moreover, it has been speculated that an ATG16L1 defect affects the role of autophagy machinery in various signaling pathways involved in modulating cytokine production, resulting in the accumulation of dysfunctional proteins and organelles, and subsequent chronic inflammation [31]. Additionally, Lee et al. reported that autophagy deficiency in keratinocytes caused increased production of inflammatory cytokines, and cell proliferation via induction of the scaffolding adaptor protein p62/SQSTM1 (p62) expression [32]. However, studies on the role of autophagy in psoriasis pathogenesis remain limited. Both AhR signaling and autophagy act as skin homeostatic rheostats sensing environmental stimuli, and evidence suggests that AhR signaling and autophagy are involved in the pathogenesis of chronic inflammatory skin disease, such as psoriasis. Thus, we suggested that AhR activation by environmental pollutants such as dioxin (TCDD) could affect the autophagic process, and might contribute to the pathogenesis of chronic inflammatory skin disorder, including psoriasis.”

  • Sukseree, S.; Eckhart, L.; Tschachler, E.; Watanapokasin, R. Autophagy in epithelial homeostasis and defense. Front. Biosci. (Elite Ed). 2013, 5, 1000-1010.
  • Douroudis, K.; Kingo, K.; Traks, T., Reimann, E.; Raud, K.; Ratsep, R.; Mössner R.; Silm H.; Vasar E.; Kõks S. Polymorphisms in the ATG16L1 gene are associated with psoriasis vulgaris. Acta Derm. Venereol. 2012, 92, 85-87.
  • Mizushima, N.; Kuma, A.; Kobayashi, Y.; Yamamoto, A.; Matsubae, M.; Takao, T.; Natsume, T.; Ohsumi, Y.; Yoshimori, T. Mouse Apg16L, a novel WD-repeat protein, targets to the autophagic isolation membrane with the Apg12-Apg5 conjugate. J Cell Sci. 2003, 116, 1679–1688.
  • Varshney, P.; Saini, N. PI3K/AKT/mTOR activation and autophagy inhibition plays a key role in increased cholesterol during IL-17A mediated inflammatory response in psoriasis. Biochim. Biophys. Acta Mol. Basis Dis. 2018, 1864, 1795-1803.

Point 30. Line 306: Please add references.

Response 30. We have added a relevant reference in line 306.

Line 406: Both AhR signaling and autophagy act as skin homeostatic rheostats sensing environmental stimuli, and evidence has been reported that AhR signaling and autophagy are involved in the pathogenesis of chronic inflammatory skin disease, such as psoriasis [21,30,32-34].

  • Quintana, F.J.; Basso, A.S.;Iglesias, A.H.; Korn, T.; Farez, M.F.; Bettelli, E.; Caccamo, M. Oukka, M.; Weiner, H.L. Control of T(reg) and T(H)17 cell differentiation by the aryl hydrocarbon receptor. Nature. 2008, 453, 65-71.
  • Douroudis, K.; Kingo, K.; Traks, T., Reimann, E.; Raud, K.; Ratsep, R.; Mössner R.; Silm H.; Vasar E.; Kõks S. Polymorphisms in the ATG16L1 gene are associated with psoriasis vulgaris. Acta Derm. Venereol. 2012, 92, 85-87.
  • Lee, H.M.; Shin, D.M.; Yuk, J.M.; Shi, G.; Choi, D.K.; Lee, S.H.; Huang, S.M.; Kim, J.M.; Kim, C.D.; Lee, J.H.; et al. Autophagy negatively regulates keratinocyte inflammatory responses via scaffolding protein p62/SQSTM1. J. Immunol. 2011, 186, 1248-1258.
  • Varshney, P.; Saini, N. PI3K/AKT/mTOR activation and autophagy inhibition plays a key role in increased cholesterol during IL-17A mediated inflammatory response in psoriasis. Biochim. Biophys. Acta Mol. Basis Dis. 2018, 1864, 1795-1803.
  • Di Meglio, P.; Duarte, J.H.; Ahlfors, H.; Owens, N.D.; Li, Y.; Villanova, F.; Tosi I.; Hirota, K.; Nestle, F.O.; Mrowietz, U.; et al. Activation of the aryl hydrocarbon receptor dampens the severity of inflammatory skin conditions. Immunity. 2014, 40, 989-1001.

Point 31. Line 306: Please rephrase this sentence: “Thus, our study suggested hypothesized that AhR activation by environmental pollutants such as dioxin (TCDD)…”

Response 31. As per your suggestions, we have corrected the sentence (line 407) as “Thus, our study suggested that AhR activation by environmental pollutants such as dioxin (TCDD)…”.

Point 32. Line 378: Why choose 10ng/ml for each cytokine in the mixture?

Response 32. We have chosen 10 ng/ml for each cytokine according to a previously published study that also included an in vitro psoriasis model. We have added the references in line 491.

16.Guilloteau, K.; Paris, I.; Pedretti, N.; Boniface, K.; Juchaux, F.; Huguier, V.; Guillet, G.; Bernard, F.X.; Lecron, J.C.; Morel, F. Skin Inflammation Induced by the Synergistic Action of IL-17A, IL-22, Oncostatin M, IL-1α, and TNF-α Recapitulates Some Features of Psoriasis. J Immunol. 2010, 184, 5263-5270.

Point 33. Line 376: “… fetal bovine serum (FBS; Gibco, Thermo Fisher Scientific, USA), …”

Response 33. In line 487, we have corrected as per your specific instructions “… fetal bovine serum (FBS; Gibco, Thermo Fisher Scientific, USA), …”.

Point 34. Lines 378, 385, 392: Please change CO2 for CO2.

Response 34. In lines 489,497,504, we have changed CO2 to CO2.

Point 35. Line 396: please change “… density of 2×104/6-well plates…” for “… density of 2×104/well in a 6-well plate…”

Response 35. As the reviewer pointed, we have changed “… density of 2×104/6-well plates…” for “… density of 2×104/well in a 6-well plate…” in line 508.

Point 36. Line 401: Does the test was optimized before because the final concentration of MTT solution is a little high and this can explain why you observed a decrease in the cell viability. Usually, a concentration of 0.5mg/ml (for 1 to 4 hours) is used in those assays.

Response 36. We apologize for our inaccurate description of the MTT concentration. As you noted, we performed the MTT assay using 0.5 mg/ml MTT solution. we have corrected the description regarding the concentration of MTT solution (line 514).

Point 37. Line 486: Please update this list with all the abbreviations (Rapa, CQ, FBS, FICZ, …, are missing).

Response 37. We have added Rapa, CQ, FBS, FICZ, EDTA, DMEM, ECL, DMSO, TBST, PBS, FITC, SDS-PAGE, GAPDH, IL-1β, IL-6 and TNF- α in the abbreviations (line 607).

Point 38. Figures in general: There is no need to use two types of identification for every condition. For an example (figure 2a) you use Control, TCDD, TCDD+M5 and you also use it with + and – on top of the first identification method. It is a little repetitive. 

Response 38. Thank you for helpful comment. We have removed + and – in figures in general.

Reviewer 4 Report

In the Article “Role of Aryl Hydrocarbon Receptor Activation and Autophagy in Psoriasis-related Inflammation” the authors addressed the role of the AhR and autophagy in an in vitro Psoriasis-model. They additionally investigated, if AhR-signaling and autophagy influence each other during inflammation and correlated their findings with psoriasis patient data.

The authors show that AhR-activation via TCDD induced inflammatory activity of human keratinocytes and inhibition of autophagy enhanced inflammation. Congruently, a higher AhR-activity and a lower amount of autophagic components in terms of LC3 in keratinocytes was found in psoriasis patients. Using autophagy-inhibitor chlorochine and -inducer rapamycin the authors analyzed a connection between AhR-activity and autophagy and present data and show that AhR-stimulation might inhibit autophagy and autophagy might decrease AhR-activity.

The chosen topic is highly interesting and the effect that TCDD reduces the formation of autophagosomes seems to consistent and is worth to be further analyzed for the underlying mechanisms. However, the study as such contains major experimental ambiguities and rational inconsistencies. Even though, the authors emphasized the limitations of the study, the data provide a confusing and doubtful information content, which I would not recommend for publication in this form.

Comments to the authors:

Major:

  • From a viability test in HaCaT-cells concentrations of TCDD, rapamycin and chloroquine were chosen, which had slight but statistical significant toxic effects on HaCaT-cells. Later these substances were applied to NHEK. For NHEK the toxicitiy of these substances was not assessed. The toxicity was also not tested for keratinocytes under the influence of inflammatory cytokines, which might alter the threshold for toxicity. Therefore, it is not excluded that observed effects result from damage in the cells.

  • No explanation was given for how rapamycin induces or chloroquine inhibits autophagy and why particularly these substances were chosen. Both substances might have several other effects, which might interfere with AhR-signaling or inflammatory activity independently from autophagy. Moreover, rapamycin did not induce LC3 in HaCat cells, which was not commented by the authors. Rapamycin did not have any effect on AhR- or Cyp1a1-protein in HaCat cells (Figure 3. c-f and Figure 4 a, b, c), but the authors claimed in the discussion that this was the case. In another set of experiments (Figure 5) rapamycin seemed to have an effect on AhR and Cyp1a1 expression with and without siAhR, but no test for significance was performed. Instead the expression levels of AhR, Cyp1a1 was compared between controls and siAhR-treated, which only demonstrates the direct consequence of AhR knockdown and does not adress effects due to altered autophagy. Despite that, the authors concluded that siAhR attenuates the rapamycin induced suppression of AhR.

  • The rationality behind the experimental setup of Figure 5 and the conclusion are not comprehensible. The knockdown of AhR and ATG5 and the statistical analysis do not give more information about the mechanism. If the authors wanted to show that an induction of autophagy dampens AhR-expression dependent on the presence of autophagosomes they should have used siATG5 in combination with rapamycin, and if they wanted to show that TCDD inhibits the formation of autophagosomes AhR-dependently they should have used TCDD and siAhR. The data present only that rapamycin probably reduced AhR expression with or without a simultaneous Ahr-knockdown and TCDD seemed to reduce autophagosomes with or without knockdown of autophagosomes.

  • In Figure 7 it is not stated how long the cells were stimulated by M5 before transfection.

  • In Material and Methods it is described that patient samples were taken from 6 healthy controls, 6 psoriasis patients and 3 AD-patients. In the figures there are also data from non-psoriasis patients and for each panel n = 3 is stated. It is not clear, if the same patients were used double or not for different experiments.

  • The conclusion of Figure 7 is unclear. The meaning of the sentence line 218 “AhR silencing was reduced in TCDD or choroquine-induced phosphorylation of p65NF and p38MAPK” is not clear. On the one hand the authors seem to claim that AhR-activation stimulates inflammation, on the other hand Figure 7 shows that AhR-silencing reduces phosphorylation of p65 and p38 without any additional stimulus and don’t comment on it.

  • A sample size of n = 3 too is small for a test for normal distribution and therefore not suitable for any statistical test. Sample sizes should be increased to at least 5 or statistics should be removed. The statistical test for calculation of significance should have been stated for each figure.

Minor:

  • Bar graphs hide important information about sample size and variation. Individual values should be shown instead.
  • Figure 5 last panel should be labeled as ‘I’
  • In Figure 6a the total protein of p65 and p38 should be shown as control instead of Gapdh
  • In Figure 7 a-h all labelings are switched, e.g. the western blot is labeled with TCDD but the + is made for M5
  • There are no comprehensive explanations given for chosen stimulation or treatment times.
  • There are wrong conclusions stated in the discussion, e.g. line 321 “inhibition of autophagy by chloroquine significantly decreased the production of TNF-alpha in NHEKs” (Figure 4d tells the opposite).
  • Figure 10: the authors claim that IL-1beta, IL-6 and TNF-alpha were significantly enhanced by TCDD and chloroquine, but the significance test does reflect this conclusion.
  • In general the results are poorly described and the reader is not well guided through the figures

Author Response

Response to Reviewer 4 Comments

In the Article “Role of Aryl Hydrocarbon Receptor Activation and Autophagy in Psoriasis-related Inflammation” the authors addressed the role of the AhR and autophagy in an in vitro Psoriasis-model. They additionally investigated, if AhR-signaling and autophagy influence each other during inflammation and correlated their findings with psoriasis patient data.

The authors show that AhR-activation via TCDD induced inflammatory activity of human keratinocytes and inhibition of autophagy enhanced inflammation. Congruently, a higher AhR-activity and a lower amount of autophagic components in terms of LC3 in keratinocytes was found in psoriasis patients. Using autophagy-inhibitor chlorochine and -inducer rapamycin the authors analyzed a connection between AhR-activity and autophagy and present data and show that AhR-stimulation might inhibit autophagy and autophagy might decrease AhR-activity.

The chosen topic is highly interesting and the effect that TCDD reduces the formation of autophagosomes seems to consistent and is worth to be further analyzed for the underlying mechanisms. However, the study as such contains major experimental ambiguities and rational inconsistencies. Even though, the authors emphasized the limitations of the study, the data provide a confusing and doubtful information content, which I would not recommend for publication in this form.

 We appreciate the insightful comments on our manuscript. Please find our point-by-point responses to each comment. We hope that the manuscript is now suitable for publication in the International Journal of Molecular Sciences. We thank you for your consideration and look forward to your response.

Comments to the authors:

Major:

Point 1. From a viability test in HaCaT-cells concentrations of TCDD, rapamycin and chloroquine were chosen, which had slight but statistical significant toxic effects on HaCaT-cells. Later these substances were applied to NHEK. For NHEK the toxicitiy of these substances was not assessed. The toxicity was also not tested for keratinocytes under the influence of inflammatory cytokines, which might alter the threshold for toxicity. Therefore, it is not excluded that observed effects result from damage in the cells.

Response 1. We assessed the toxicity of TCDD, rapamycin and chloroquine in NHEK. Under the chosen concentrations, cell viability was consistently higher than 90%. As you pointed out, in Fig. 1, there are significant decreases in comparison to unstimulated cell in HaCaT cells. However, the cell viability in the chosen concentrations were all over 90% in HaCaT cells, which is within the generally accepted range. In previous studies using TCDD, rapamycin and chloroquine, the chosen concentrations were also used.

TCDD

1)Aryl Hydrocarbon Receptor Modulates the Expression of TNF-α and IL-8 in Human Sebocytes via the MyD88-p65NF-κB/p38MAPK Signaling Pathways. J Innate Immun. 2019;11(1):41-51.

2) 2,3,7,8-Tetrachlorodibenzo-p-dioxin Increases the Expression of Genes in the Human Epidermal Differentiation Complex and Accelerates Epidermal Barrier Formation. Toxicol Sci. 2011 Nov;124(1):128-37

Chloroquine

1) Chloroquine treatment of ARPE-19 cells leads to lysosome dilation and intracellular lipid accumulation: Possible implications of lysosomal dysfunction in macular degeneration. Cell Biosci. 2011 Mar 8;1(1):10.

2) Quercetin induces protective autophagy in gastric cancer cells: Involvement of Akt-mTOR- and hypoxia-induced factor 1α-mediated signaling. Autophagy. 2011 Sep;7(9):966-78.

Rapamycin

Synergistic Inhibition of Tumor Necrosis Factor-Alpha-Stimulated Pro-Inflammatory Cytokine Expression in HaCaT Cells by a Combination of Rapamycin and Mycophenolic Acid. Ann Dermatol. 2015 Feb;27(1):32-9.

We are sorry that we were unable to perform the test to evaluate the toxicity of these substances under the influence of inflammatory cytokines (M5) due to time limitation for revision. However, the chosen concentration was selected according to that presented in a previously published study on M5 (16).

  • Guilloteau, K.; Paris, I.; Pedretti, N.; Boniface, K.; Juchaux, F.; Huguier, V.; Guillet, G.; Bernard, F.X.; Lecron, J.C.; Morel, F. Skin Inflammation Induced by the Synergistic Action of IL-17A, IL-22, Oncostatin M, IL-1α, and TNF-α Recapitulates Some Features of Psoriasis. J Immunol. 2010, 184, 5263-5270.

Point 2. No explanation was given for how rapamycin induces or chloroquine inhibits autophagy and why particularly these substances were chosen. Both substances might have several other effects, which might interfere with AhR-signaling or inflammatory activity independently from autophagy.

Response 2. As you pointed out, both substances may have several other effects independent from autophagy. Besides its direct inhibitory effect on autophagy, chloroquine can also activate autophagy-independent NOTCH1 signaling, resulting in reduced hypoxia in cancer cells (Cancer Cell 2014;26:190–206). Furthermore, rapamycin has been shown to elicit immunosuppressive and anti-proliferative properties, and has been widely used for treatment of refractory acute rejection and neurodegenerative diseases (Redox Biol 2014;2:82-90). Nevertheless, among the many autophagy-modulating drugs, chloroquine, an antimalarial drug, is a representative autophagy inhibitor that blocks the degradation of autophagosomes; while rapamycin is regarded as an autophagy inducer and inhibits mammalian TOR (mTOR), which is serine/threoine kinase. Since autophagy is regulated by the mTOR complex, rapamycin can induce autophagy via inhibition of mTOR. Therefore, we selected rapamycin as an autophagy inducer and chloroquine as an autophagy inhibitor. This description was added in result section (line, 154-159).

Point 3. Moreover, rapamycin did not induce LC3 in HaCat cells, which was not commented by the authors. Rapamycin did not have any effect on AhR- or Cyp1a1-protein in HaCat cells (Figure 3. c-f and Figure 4 a, b, c), but the authors claimed in the discussion that this was the case.

Response 3. Thank you for helpful comments. First of all, we have removed the bar graphs in Fig. 3d-f, as per another reviewer’s suggestion (minor point 10). Secondly, we have presented individual densitometry values for representative data of western analysis (Results are representative of three independent experiments.). In Fig. 3a and 3b, rapamycin did not induce AHR and CYP1A1 mRNA expression. However, a significant downregulation of CYP1A1 mRNAs was shown in rapamycin + M5-treated HaCaT cells. In western blot analysis (Fig. 3c), as the individual values of densitometry, rapamycin lead to a slight decrease in the level of AhR and CYP1A1 proteins compared to control. LC3 protein level was slightly increased by rapamycin compared with control. We agree that rapamycin exhibited no significant effects on AhR- , CYP1A1- or LC3 expression in HaCat cells. However, we found that rapamycin induced a significant downregulation of CYP1A1 mRNAs in M5-treated HaCaT cells (Fig.3B). To confirm the effects of rapamycin on AhR, CYP1A1, and LC expression, we further performed immunofluorescence staining in NHEKs and found that rapamycin did not have an effect on AhR or CYP1A1 protein expression. However, rapamycin showed significant induction of LC3 protein in NHEKs. These different effects of rapamycin on LC3 protein induction may be caused by different cell specific effects. In this study, the effect of autophagy induction by rapamycin was only confirmed by changes in LC3 protein expression. Hence, to further assess the definitive effect of rapamycin as an autophagy inducer, the other measurements of autophagic flux using western blotting with other autophagy-related factors (ATG5, Beclin1, p62, etc.) and IF or transmission electron microscopy analysis to evaluate autophagosome formation, may be needed. we have added these descriptions regarding in discussion section. (line 450-457).

Point 4. In another set of experiments (Figure 5) rapamycin seemed to have an effect on AhR and Cyp1a1 expression with and without siAhR, but no test for significance was performed. Instead the expression levels of AhR, Cyp1a1 was compared between controls and siAhR-treated, which only demonstrates the direct consequence of AhR knockdown and does not adress effects due to altered autophagy. Despite that, the authors concluded that siAhR attenuates the rapamycin induced suppression of AhR.

Response 4. We have rechecked the results of Fig 5 and found that rapamycin did not have a significant effect on AhR or Cyp1a1 expression. Thus, the effect of rapamycin and other autophagy inducers will need to be examined further on AhR or CYP1A1. The original Fig. 5 (B,C) demonstrated only the direct consequence of AhR knockdown and was unable to address effects due to altered autophagy. Therefore, we have deleted the previous Fig. 5 (B, C) regarding rapamycin until further confirmation of these results can be obtained. We apologize that we were unable to perform these confirmatory tests now due to short revision time allotment.

Point 5. The rationality behind the experimental setup of Figure 5 and the conclusion are not comprehensible. The knockdown of AhR and ATG5 and the statistical analysis do not give more information about the mechanism. If the authors wanted to show that an induction of autophagy dampens AhR-expression dependent on the presence of autophagosomes they should have used siATG5 in combination with rapamycin, and if they wanted to show that TCDD inhibits the formation of autophagosomes AhR-dependently they should have used TCDD and siAhR. The data present only that rapamycin probably reduced AhR expression with or without a simultaneous Ahr-knockdown and TCDD seemed to reduce autophagosomes with or without knockdown of autophagosomes.:

Response 5. As per your recommendation, we repeated the experiment using TCDD and siAhR to confirm that TCDD inhibits the formation of autophagosomes AhR-dependently. Unfortunately, due to the explanation provided in response 3 and 4, the effect of rapamycin on AhR, CYP1A1 and LC3 protein expression was unclear. Thus, we were unable to do the experiment with siATG5 in combination with rapamycin. In the revised Fig.5, AhR-knockdown showed more inhibition of TCDD-induced attenuation of autophagy-related protein expression and autophagosome production. Therefore, taken together, TCDD inhibited autophagy AhR-dependently.(line 229-237)

Point 6. In Figure 7 it is not stated how long the cells were stimulated by M5 before transfection.

Response 6. In Fig. 7 (Fig. 6 in revised manuscript). HaCaT cells were stimulated by M5 for 48hr before transfection. We have added the stimulation duration of M5 in the legend for Fig. 7 (Fig. 6 in revised manuscript).

Point 7. In Material and Methods it is described that patient samples were taken from 6 healthy controls, 6 psoriasis patients and 3 AD-patients. In the figures there are also data from non-psoriasis patients and for each panel n = 3 is stated. It is not clear, if the same patients were used double or not for different experiments.

Response 7. We apologize for not clearly describing our patient groups. In IHC and IF, the same subject specimens were used (psoriasis patient = 3, healthy control = 3). However, we have deleted IF results in Fig. 8 due to repetitive data with IHC. For qPCR (Fig.8 in revised manuscript) and the ex vivo study using skin biopsy tissues (Fig.9 in revised manuscript) were performed on the same three psoriasis patients and three normal controls. qPCR (Fig.8 in revised manuscript ) and ex vivo study (Fig.9 in revised manuscript were different experiments than the IHC analysis. Therefore, in Material and Methods section it is described that patient samples were taken from 6 healthy controls, 6 psoriasis patients (line 480-481).

Point 8. The conclusion of Figure 7 is unclear. The meaning of the sentence line 218 “AhR silencing was reduced in TCDD or choroquine-induced phosphorylation of p65NF and p38MAPK” is not clear. On the one hand the authors seem to claim that AhR-activation stimulates inflammation, on the other hand Figure 7 shows that AhR-silencing reduces phosphorylation of p65 and p38 without any additional stimulus and don’t comment on it.

Response 8. First of all, we have removed bar graphs from Fig. 7 (Fig.6 in revised manuscript) according to your minor point 10. We have instead presented individual densitometry values of representative data (Results are representative of three independent experiments.). We have described the results (line 285-291) for densitometry values in Fig. 7 (Fig.6 in revised manuscript). “As shown in Figure 6a and 6b, TCDD treatment caused a slight increase in the level of phospho-p38 (p-p38) and p-p65, similar to M5, and the M5/TCDD combination induced a stronger effect compared to the single treatments. Chloroquine induced a slight increase in the phosphorylation of p38 and p65. Similarly, M5 lead to an increase in the level of p-p38 and p-p65. The combination of M5 and chloroquine also induced a stronger effect compared to the individual treatments (Fig. 6c, d). AhR silencing was reduced in TCDD or chloroquine-induced phosphorylation of p65NF-κB and p38MAPK (Fig. 6a-d).

Point 9. A sample size of n = 3 too is small for a test for normal distribution and therefore not suitable for any statistical test. Sample sizes should be increased to at least 5 or statistics should be removed. The statistical test for calculation of significance should have been stated for each figure.

Response 9.

In the original Fig. 8, IHC analysis was performed for three healthy controls, and three psoriasis patients. Thus, we have deleted the results of statistical test and bar graphs according to your comment. In western blot analysis, we have also removed bar graphs and presented the representative data of three independent experiments with individual values of densitometric analysis.

We have added the statistical test for calculation of significance in each figure legends.

Minor:

Point 10. Bar graphs hide important information about sample size and variation. Individual values should be shown instead.

Response 10. As the reviewer mentioned, we have removed the unnecessary bar graphs. Instead we have described the individual values of densitometry measurements in all western blot analysis.

Point 11. Figure 5 last panel should be labeled as ‘I’

Response 11. In Fig. 5, we have replaced new results of TCDD/ siAhR experiment. Therefore, previous Fig. 5I was removed.

Point 12. In Figure 6a the total protein of p65 and p38 should be shown as control instead of Gapdh

Response 12. In previous Fig. 7, we have used the total proteins of p65 and p38 as control. However, as you pointed out, GAPDH was used as a control in previous Fig. 6. We agree with your comment. However, due to short time limitation of revision set by IJMS, we were unable to complete additional experiments to supplement Fig. 6 using the total protein of p65 and p38. Therefore, we replaced previous Fig. 6 as supplementary material Fig. 1. In our future work we will use the total proteins of p65 and p38 as control.

Point 13. In Figure 7 a-h all labelings are switched, e.g. the western blot is labeled with TCDD but the + is made for M5

Response 13. Thank you for your comment. We apologize for this oversight. We have deleted +, - labels due to repetitive information.

Point 14. There are no comprehensive explanations given for chosen stimulation or treatment times.

Response 14. We have chosen stimulations or treatment times of substances according to those described in previously published studies:

TCDD

1)Aryl Hydrocarbon Receptor Modulates the Expression of TNF-α and IL-8 in Human Sebocytes via the MyD88-p65NF-κB/p38MAPK Signaling Pathways. J Innate Immun. 2019;11(1):41-51.

2) 2,3,7,8-Tetrachlorodibenzo-p-dioxin Increases the Expression of Genes in the Human Epidermal Differentiation Complex and Accelerates Epidermal Barrier Formation. Toxicol Sci. 2011 Nov;124(1):128-37

M5

IL-37 ameliorates the inflammatory process in psoriasis by suppressing proinflammatory cytokine production. J Immunol. 2014 Feb 15;192(4):1815-23.

Chloroquine

1) Chloroquine treatment of ARPE-19 cells leads to lysosome dilation and intracellular lipid accumulation: Possible implications of lysosomal dysfunction in macular degeneration. Cell Biosci. 2011 Mar 8;1(1):10.

2) Quercetin induces protective autophagy in gastric cancer cells: Involvement of Akt-mTOR- and hypoxia-induced factor 1α-mediated signaling. Autophagy. 2011 Sep;7(9):966-78.

Rapamycin

Synergistic Inhibition of Tumor Necrosis Factor-Alpha-Stimulated Pro-Inflammatory Cytokine Expression in HaCaT Cells by a Combination of Rapamycin and Mycophenolic Acid. Ann Dermatol. 2015 Feb;27(1):32-9.

Point 15. There are wrong conclusions stated in the discussion, e.g. line 321 “inhibition of autophagy by chloroquine significantly decreased the production of TNF-alpha in NHEKs” (Figure 4d tells the opposite).

Response 15. We appreciate your point. Accordingly, we have revised the sentence to: ‘inhibition of autophagy by chloroquine significantly increased the production of TNF-alpha in NHEKs.” (line 421)

Point 16. Figure 10: the authors claim that IL-1beta, IL-6 and TNF-alpha were significantly enhanced by TCDD and chloroquine, but the significance test does reflect this conclusion.

Response 16. As per your comment we rechecked the results in Fig. 10 (Fig. 9 in revised manuscript) and found that IL-1beta, IL-6 and TNF-alpha were not significantly enhanced by TCDD and chloroquine. In Fig. 10 (Fig. 9 in revised manuscript), we found that in psoriasis skin biopsies, TCDD and CQ significantly enhanced the expression of IL-1β, IL-6, and TNF-α compared to control skin biopsies. We have corrected the sentence, “Chloroquine or TCDD treatment of psoriasis skin biopsies enhanced the production of the proinflammatory cytokines, IL-1β, IL-6, and TNF-α” as “Chloroquine or TCDD treatment of psoriasis skin biopsies enhanced the production of the proinflammatory cytokines, IL-1β, IL-6, and TNF-α compared to chloroquine- or TCDD- treated controls.” (line 320-322)

Point 17. In general the results are poorly described and the reader is not well guided through the figures

Response 17. Through the figures, we have corrected the results which were poorly described.

Round 2

Reviewer 4 Report

The authors have significantly improved the content of the manuscript. Major inconsistencies have been removed and the results are presented more comprehensible. However, some points should still be revised before publication.

  1. Conclusions driven from figure 5 are not supported by the presented data:
    1. In line 194 it is claimed: 'AhR knockdown resulted in significant attenuation of TCDD-induced reduced expression of autophagy'
    2. However, white bars in Figure 5 b,c,e clearly show that TCDD induces ATG5, LC3, and Beclin1 (even if there was no statistical test performed). True is that siAhR reduce these factors
    3. The main conclusion of this paragraph should be changed to e.g.'AhR knockdown resulted in significant attenuation of TCDD-induced expression of autophagy'. And an ANOVA with Tukey's test comparing Mock to TCDD, M5 and TCDD+M5 should be performed.
  2. Line 232 ‘AhR silencing was reduced following TCDD or chloroquine-induced phosphorylation of p65NF-kB and p38MAPK’ makes no sense and should be revised to ‘AhR silencing reduced phosphorylation of p65NF-kB and p38MAPK independently of M5, TCDD or CQ-treatment.’

  3. The densiometric values do partly not reflect the relative thicknesses of bars. E.g. in Figure 6. b, c, d Mock-control has the value 1.0 and siAhR-control is labelled with 0.51 or 0.53 despite of a clearly visible thicker band in siRNA-control. In general, siAhR seems to not only attenuate the TCDD or CQ-effect, but results in even stronger reduction p65 and p38-phosphorylation independently of TCDD or CQ. (compare M5-Control and M5-siAhR Fig. 6 a-d) Authors should comment on both phenomenons and discuss how a loss of AhR-activity might impair p65 or p38 phosphorylation independently of autophagy.

  4. Figure 6E: One lonely star too much
  5. Line 260: missing space

Author Response

Response to Reviewer Comments

The authors have significantly improved the content of the manuscript. Major inconsistencies have been removed and the results are presented more comprehensible. However, some points should still be revised before publication.

Point 1. Conclusions driven from figure 5 are not supported by the presented data:
a. In line 194 it is claimed: 'AhR knockdown resulted in significant attenuation of TCDD-induced reduced expression of autophagy'
b. However, white bars in Figure 5 b,c,e clearly show that TCDD induces ATG5, LC3, and Beclin1 (even if there was no statistical test performed). True is that siAhR reduce these factors
c. The main conclusion of this paragraph should be changed to e.g.'AhR knockdown resulted in significant attenuation of TCDD-induced expression of autophagy'. And an ANOVA with Tukey's test comparing Mock to TCDD, M5 and TCDD+M5 should be performed.

Response 1. We appreciate your helpful comment. We agree that TCDD induced ATG5, LC3, and Beclin1 expression (However, we found that there were no significant differences compared to control levels based on an ANOVA with Tukey’s test, comparing Mock to TCDD, M5, and TCDD+M5 groups); Moreover, AhR silencing reduced the expression of these factors. Based on your comment and these results, we presume that TCDD or M5 did not significantly affect the mRNA levels of ATG5, LC3, and Beclin1. Furthermore, AhR knockdown reduced the expression of autophagy-related factors relative to that in the respective controls. We have, therefore, corrected this statement to read as follows: “AhR knockdown resulted in significant attenuation of the expression of autophagy-related factors relative to that in the respective controls.” (line 226-227 with green highlight)

Point 2. Line 232 ‘AhR silencing was reduced following TCDD or chloroquine-induced phosphorylation of p65NF-kB and p38MAPK’ makes no sense and should be revised to ‘AhR silencing reduced phosphorylation of p65NF-kB and p38MAPK independently of M5, TCDD or CQ-treatment.’

Response 2. As per your suggestions (point 2 and 3), we have revised the sentence in line 232 to “AhR knockdown seems to not only attenuate the TCDD- or chloroquine (CQ)-mediated effects but also results in even stronger reductions in p65 and p38 phosphorylation, independently of TCDD or CQ.” (line 244-245).

Point 3. The densiometric values do partly not reflect the relative thicknesses of bars. E.g. in Figure 6. b, c, d Mock-control has the value 1.0 and siAhR-control is labelled with 0.51 or 0.53 despite of a clearly visible thicker band in siRNA-control. In general, siAhR seems to not only attenuate the TCDD or CQ-effect, but results in even stronger reduction p65 and p38-phosphorylation independently of TCDD or CQ. (compare M5-Control and M5-siAhR Fig. 6 a-d) Authors should comment on both phenomenons and discuss how a loss of AhR-activity might impair p65 or p38 phosphorylation independently of autophagy.

Response 3. The densitometric values have been measured as total p-38 (or total p-65) values minus p-p38 (or p-p65) values. Therefore, the densitometric values are now shown as per your suggestions. We have also added a description of how a loss of AhR activity might impair p65 or p38 phosphorylation independently of autophagy as follows.
In present study, the loss of AhR activity might impair p65 or p38 phosphorylation in an autophagy-independent manner as well as in an autophagy-dependent manner. AhR has various ligands and different actions according to its ligand. It also affects various complex responses following its stimulation or inhibition (40) Canonical and non-canonical signaling pathways activated by AhR have been discovered (19). In addition to well-known canonical AhR signaling, another pathway mediated by p65NF-kB among non-canonical AhR signaling has been identified (42). Therefore, one single mechanism is not sufficient to explain the functions of AhR. Thus, more studies are required to unravel the functions and signaling pathways related to AhR.
These descriptions have added to the discussion section of revised manuscript (line 376-383).

19.Esser, C.; Rannug, A. The aryl hydrocarbon receptor in barrier organ physiology, immunology, and toxicology. Pharmacol. Rev. 2015, 67, 259-279.
40.Haarmann-Stemmann, T.; Esser, C.; Krutmann, J. The Janus-Faced Role of Aryl Hydrocarbon Receptor Signaling in the Skin: Consequences for Prevention and Treatment of Skin Disorders. J. Invest. Dermatol. 2015, 135, 2572-2576.
41.Stobbe-Maicherski, N.; Wolff, S.; Wolff, C.; Abel, J.; Sydlik, U.; Frauenstein, K.; Haarmann-Stemmann, T. The interleukin-6-type cytokine oncostatin M induces aryl hydrocarbon receptor expression in a STAT3-dependent manner in human HepG2 hepatoma cells. FEBS J. 2013, 280, 6681-6690.

Point 4. Figure 6E: One lonely star too much

Response 4. Thank you for your valuable point. In Fig. 6, there was no panel E. Therefore, we presume that you are referring to Fig. 9E. We have corrected the asterisks in Fig. 9E (the asterisk in the lowest space: non-treated control vs. non-treated psoriasis, middle asterisk: CQ-treated control vs. CQ-treated psoriasis, upper asterisk: TCDD-treated control vs. TCDD-treated psoriasis).

Point 5. Line 260: missing space

Response 5. We have identified this missing space in line 260 and corrected this. (line 270 with green highlight in revised manuscript)